# Privacy-Preserving Energy-Based Generative Models for Marginal Distribution Protection

**Robert E. Tillman**[*]
*Optum AI Labs (UnitedHealth Group)*

**Tucker Balch**
*J.P. Morgan Chase AI Research*

**Manuela Veloso**
*J.P. Morgan Chase AI Research*

**Reviewed on OpenReview:** *https://openreview.net/forum?id=vTsfup5ll6*

## Abstract

We consider learning generative models for sensitive financial and healthcare data. While previous work incorporates Differential Privacy (DP) into GAN training to protect the privacy of individual training instances, we consider a different privacy context where the primary objective is protecting the privacy of sensitive marginal distributions of the true generative process. We propose and motivate a new notion of privacy: $\alpha$-*Level Marginal Distribution Privacy* ($\alpha$-LMDP), which provides a statistical guarantee that the sensitive generative marginal distributions are different from the observed real data. We then propose *Privacy-Preserving Energy Models (PPEMs)*, a novel energy-based generative model formulation where the representations for these attributes are isolated from other attributes. This structured formulation motivates a learning procedure where a penalty based on a statistical goodness of fit test, the *Kernel Stein Discrepancy*, can be applied to only the attributes requiring privacy so that $\alpha$-LMDP may be satisfied without affecting the other attributes. We evaluate this approach using financial and healthcare datasets and demonstrate that the resulting learnt generative models produce high fidelity synthetic data while preserving privacy. We also show that PPEMs can incorporate both $\alpha$-LMDP *and* DP in contexts where both forms of privacy are required.

## 1 Introduction

In recent years, the quantity and variety of data collected from individuals and organizations has grown dramatically. Such growth has fueled the development of large-scale machine learning systems capable of high accuracy predictions, but also led to significant concerns about data privacy and the potential leakage of sensitive information. These concerns are particularly relevant for the financial and healthcare industries as their data are often very sensitive and personally identifiable and these industries are subject to strict regulatory and compliance requirements. As a result, data access at such organizations is typically highly controlled and the usage of data for research purposes often requires lengthy legal reviews with no guarantees of success (Choi et al., 2017). These restrictions often apply not only to external researchers, but to internal research groups working in different parts of the same company (Assefa et al., 2020). Ultimately, these processes can impede opportunities for academic collaborations and potential advances in areas such as financial fraud detection, anti-money laundering, disease prediction and patient care (Nass et al., 2009).

Advances in neural generative models such as GANs (Goodfellow et al., 2014) have led many such organizations to consider using synthetic data to enable more timely and collaborative research. Recent work, however,

---

[*]This work was completed at J.P. Morgan Chase AI Research.

has shown that GANs may still leak sensitive information: GAN attack models have been proposed for inferring membership in or reconstructing training data (Shokri et al., 2017; Hayes et al., 2019). This led to the development of GANs incorporating *Differential Privacy (DP)* to mitigate such attacks (Xie et al., 2018; Yoon et al., 2019). DP (Dwork & Roth, 2014) is a paradigm for protecting the privacy of training data by ensuring a learning algorithm's output does not vary significantly based on the inclusion of a particular training instance. While DP is effective for ensuring the privacy of particular individuals or organizations is protected, it does not, however, protect against the potential leakage of aggregate-level information which may be inferred from the generative distribution. This is a particular concern for (e.g.) financial institutions as generative models for financial data may leak aggregate-level information about typical client characteristics or markets the institution is most active in. We thus propose a new notion of privacy which may be used either in place of *or* in conjunction with DP, depending on the privacy requirements, and a robust procedure for learning generative models while satisfying this privacy notion.

## 1.1 Contributions

We make the following novel contributions:

1. We propose a new notion of privacy for protecting sensitive marginal distributions of the generative process: $\alpha$-*Level Marginal Distribution Privacy* ($\alpha$-*LMDP*), which provides a statistical guarantee that sensitive generative marginal distributions differ from the observed real data.

2. We theoretically show that $\alpha$-*LMDP* is neither strictly stronger nor weaker than DP.

3. We propose *Privacy-Preserving Energy Models (PPEMs)*, a novel energy-based generative model formulation where representations for attributes requiring $\alpha$-LMDP protection are isolated from other attributes.

4. We introduce a non-negative normalized variant of the *Kernel Stein Discrepancy*, a goodness-of-fit test that is compatible with energy models, which we use to incorporate $\alpha$-LMDP protection into PPEMs through a training penalty.

5. We show that when PPEM training converges, the learnt generative models produce samples indistinguishable from the real data and provide conditions under which $\alpha$-LMDP will be satisfied.

6. We show how DP can also be incorporated into PPEMs when both privacy notions are required.

7. Using credit card data and electronic healthcare records, we empirically demonstrate that PPEMs produce high fidelity synthetic data while preserving privacy.

## 2 Notions of Privacy for Generative Models

*Differential Privacy (DP)* is a paradigm for quantifying how much the output of a mechanism that interacts with data may be affected by changing a single data point in a dataset.

**Definition 2.1** (Dwork & Roth, 2014)**.** *A randomized mechanism $\mathcal{M}$ is $(\epsilon, \delta)$-Differentially Private ($(\epsilon, \delta)$-DP) if for any output set $\mathcal{S}$ and two datasets $\mathcal{D}$ and $\mathcal{D}'$ which differ by only a single data point, the following holds for $\epsilon, \delta \in \mathbb{R}^+$, where $\mathbb{P}(\cdot)$ denotes the probability taken with respect to $\mathcal{M}$:*

$$\mathbb{P}(\mathcal{M}(\mathcal{D}) \in \mathcal{S}) \leq \exp(\epsilon)\mathbb{P}(\mathcal{M}(\mathcal{D}') \in \mathcal{S}) + \delta.$$

A mechanism that is $(\epsilon, \delta)$-DP guarantees changing any data instance will affect the probability of an outcome only up to a multiplicative ($\epsilon$) and additive ($\delta$) factor, providing protection against membership inference attacks. *Gradient Perturbation* is an efficient way to augment a learning algorithm to make it DP, such as in *Differentially Private Stochastic Gradient Descent (DP-SGD)* (Abadi et al., 2016), where at each descent step, the gradient norm is clipped and Gaussian noise, which is carefully scaled to ensure a particular $(\epsilon, \delta)$-level, is added to the gradient. DP-GAN (Xie et al., 2018) is a GAN which uses DP-SGD to learn to

generate synthetic data with DP guarantees. While DP is effective for protecting the privacy of individuals and organizations whose information make up specific instances of training datasets, it does not, however, consider the privacy of aggregate-level information that may be inferred from the distribution of the generated data. For example, consider a dataset consisting of customer information from a financial institution. A DP generative model would ensure that the privacy of individual customers is not violated, but the distribution of certain attributes in the generated data might reveal information about the institution's typical customer profiles that is valuable to a competitor. Similarly, many financial transaction and trade datasets contain no personally identifiable attributes, but their marginal distributions reveal information about which markets and venues the institution is most active in. Unless this risk is managed, institutions will likely not approve of the release and use of synthetic data, limiting potential collaborations with external researchers as well as among internal teams. Similarly in healthcare, there may be ethical concerns that revealing marginal distributions of demographics associated with specific outcomes in electronic healthcare records through synthetic data could lead to discrimination in patient care or insurer coverage.

We thus propose a new statistical notion of privacy for the context of protecting sensitive marginal distributions: *$\alpha$-Level Marginal Distribution Privacy ($\alpha$-LMDP)*. $\alpha$-LMDP makes use of a *level-$\alpha$ test*, or statistical hypothesis test with probability of falsely rejecting the null hypothesis $\leq \alpha$. $\alpha$-LMDP ensures that for a specified subset of attributes whose marginal distribution may be sensitive, a level-$\alpha$ test that the training data for these attributes follows the learnt (marginal) generative distribution can be rejected.

**Definition 2.2.** *Let $x \in \mathcal{X}^d$ be a d-dimensional vector, $\{x', x^*\}$ a partitioning of the dimensions of $x$, $\mathcal{G} : \mathcal{L}^{d'} \to \mathcal{X}^d$ a generative model with output distribution $p_\mathcal{G}$, $\mathcal{D}_x = \{x_i\}_{i=1}^n$ a dataset of vectors $x \in \mathcal{X}^d$ and $\mathcal{D}_{x'} = \{x'_i\}_{i=1}^n$ the corresponding dimensions $x'$, $p_{\mathcal{G}_{x'}} = \int_{x^* \in \mathcal{X}^{d*}} p_\mathcal{G}$ the marginal generative distribution for $x'$ and $\Xi$ a level-$\alpha$ goodness of fit test. $\mathcal{G}$ satisfies $\alpha$-Level Marginal Distribution Privacy ($\alpha$-LMDP) with respect to $\Xi$ and $x'$ if the null hypothesis $H_0 : D_{x'} \sim p_{\mathcal{G}_{x'}}$ can be rejected using $\Xi$.*

The specific test utilized by $\alpha$-LMDP to provide a statistical guarantee that the privacy of the specified marginal distribution is protected is a *goodness of fit test*, which tests whether a sample fits a specified distribution. The sample corresponds to the private attributes ($x'$) in the real data and the specified distribution is the marginal distribution learnt by the generative model. The definition does not constrain the other (non-private) attributes $x^*$ as it assumes all sets of attributes whose marginal distributions are sensitive are included in $x'$. $\alpha$-LMDP requires a representation for $p_\mathcal{G}$ and a goodness of fit test capable of evaluating this representation. We propose a framework providing these requirements in section 3. We note that the definition assumes $\mathcal{G}$ and $p_\mathcal{G}$ have sufficient representation power and $\Xi$ has sufficient statistical power to reject the null hypothesis; otherwise, a training procedure soundly designed to ensure this privacy notion may still fail in practice. The framework we propose in section 3 makes use of a powerful nonparametric goodness of fit test which permits flexible representations for $p_\mathcal{G}$, the *Kernel Stein Discrepancy* (Liu et al., 2016).

An important distinction between $\alpha$-LMDP and DP is $\alpha$-LMDP is a *model-based* notion of privacy, whereas DP is *mechanism-based*. We can still, however, relate $\alpha$-LMDP to mechanisms defined to return generative models which satisfy $\alpha$-LMDP under certain conditions, as we describe in section 3. This is advantageous since mechanisms often require assumptions that are not always realistic. In such cases, the resulting models may be checked after training to ensure they satisfy $\alpha$-LMDP and accepted or retrained under different conditions depending on whether the test is rejected. When we consider $\alpha$-LMDP in regards to such mechanisms, we can formally show that it is neither a strictly weaker nor strictly stronger notion of privacy than $(\epsilon, \delta)$-DP, i.e. $(\epsilon, \delta)$-DP mechanisms which return generative models never also guarantee any level of $\alpha$-LMDP and vice-versa. While such results are expected since $\alpha$-LMDP and DP address completely different privacy concerns, we provide this theoretical analysis since, in practice, DP training procedures often affect the learnt generative distributions. We provide proofs in Appendix B.

**Theorem 2.1.** *For a generative mechanism $\mathcal{M}$, $\alpha < 1$, a partitioning $x = \{x^*, x'\}$ and $\epsilon, \delta \in \mathbb{R}^+$, $\mathcal{M}$ is $(\epsilon, \delta)$-DP $\not\Rightarrow \mathcal{M}$ returns models satisfying $\alpha$-LMDP for $x'$.*

**Theorem 2.2.** *For a generative mechanism $\mathcal{M}$ and partitioning $x = \{x^*, x'\}$, $\mathcal{M}$ returns models satisfying $\alpha$-LMDP for $x' \not\Rightarrow \mathcal{M}$ is $(\epsilon, \delta)$-DP for finite $\epsilon, \delta$.*

Since $\alpha$-LMDP and DP address different types of privacy concerns, we cannot rely on using one to entail the other. Our proposed model in section 3 is capable of incorporating both $\alpha$-LMDP and DP if required.

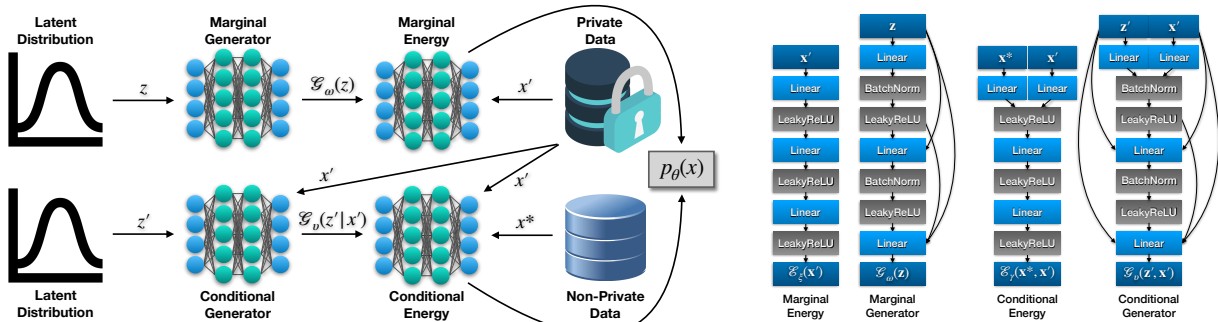

Figure 1: Diagram of PPEMs (left) and network architectures of the individual components (right)

# 3 Privacy-Preserving Energy Models (PPEMs)

We first provide background on GANs as well as energy models and then we propose *Privacy-Preserving Energy Models (PPEMs)*.

## 3.1 Background

A *Generative Adversarial Networks (GAN)* consists of a *generator* $\mathcal{G}_\phi$ parameterized by $\phi$ which maps samples $z$ from a latent distribution $p_z$ to the real distribution $p_x$ and a *discriminator* $\mathcal{D}_\theta$ parameterized by $\theta$ which attempts to distinguish real from generated samples. In Goodfellow et al. (2014), $\mathcal{G}_\phi$ and $\mathcal{D}_\theta$ are trained adversarially using the following minimax loss, which minimizes the Jensen-Shannon divergence between the real and synthetic distributions:

$$\min_\phi \max_\theta \ \left( \mathbb{E}_{p_x} \log \mathcal{D}_\theta(x) + \mathbb{E}_{p_z} \log \left[ 1 - \mathcal{D}_\theta \left( \mathcal{G}_\phi(z) \right) \right] \right) \tag{1}$$

As training with the above loss is notoriously unstable, alternative formulations have been proposed. The *Wasserstein GAN (WGAN)* (Arjovsky et al., 2017) replaces the discriminator with a *critic* that scores, rather than classifies, the *realness* of a sample. It minimize the Wasserstein-1 distance by enforcing a Lipschitz constraint on the critic and generally leads to more stable training. The original formulation was implemented using parameter clipping (Arjovsky et al., 2017). Gulrajani et al. (2017) proposes an alternative formulation which appends the below soft penalty, where $u \sim Unif(0,1)$ and $\hat{x} = ux + (1-u)\mathcal{G}_\phi(z)$, to the critic loss and generally leads to improved performance.

$$\lambda_L \left( \| \nabla_{\hat{x}} \mathcal{D}_\theta \left( \hat{x} \right) \|_2 - 1 \right)^2 \tag{2}$$

An *energy-based model (EBM)* (Saremi et al., 2018) $\mathcal{E}_\theta : \mathcal{X}^d \to \mathbb{R}^+$ parameterized by $\theta$ maps $d$-dimensional samples $x \in \mathcal{X}^d$ from an unknown distribution $p_x$ to scalar *energy* values, resulting in the following density $p_\theta$ with normalization $Z_\theta$, referred to as the *partition function*:

$$p_\theta(x) = \frac{1}{Z_\theta} \exp\left( -\mathcal{E}_\theta(x) \right) \quad Z_\theta = \int_x \exp\left( -\mathcal{E}_\theta(x) \right) \tag{3}$$

$\mathcal{E}_\theta(\cdot)$ thus represents an unnormalized log-density: lower energies correspond to higher probability events. EBMs offer considerable flexibility in the designing the energy function, often a neural network. Their primary challenge is estimating the (generally intractable) partition function.

*EnergyGAN* (Zhao et al., 2017) uses adversarial training to learn an energy model: the GAN framework is adopted, but the discriminator is replaced with an energy model and trained to promote assigning lower energies to real samples and higher energies to samples from the generator. We adopt the same framework in the model we propose, which results in not requiring evaluation of the partition function.

### 3.2 Proposed Model

We propose *Privacy-Preserving Energy Models (PPEMs)* which consist of two EBMs and two generators: one EBM and generator pair models the marginal distribution of the attributes $x'$ for which $\alpha$-LMDP must be satisfied while the other pair models the distribution of the other attributes $x^*$ conditioned on $x'$. The purpose of this structured formulation is to allow for a training procedure that ensures the resulting generators satisfy $\alpha$-LMDP with respect to $x'$ without impacting the the generated non-sensitive attributes $x^*$ and their relation to $x'$. This permits the generation of paired values for the attributes $\{x', x^*\}$ that are *individually* realistic given the real data, but *in the aggregate* do not reveal the true marginal distribution of $x'$. Figure 1 (left) provides a schema depicting how the individual PPEM components, defined below, each interact with the data.

Let $x \in \mathcal{X}^d$ be a $d$-dimensional vector and $\{x', x^*\}$ a partitioning of the dimensions of $x$ such that $x'$ corresponds to the attributes with sensitive marginal distributions that require $\alpha$-LMDP protection. Let $d'$ and $d^*$ be the corresponding dimensions of $x'$ and $x^*$. We define the *marginal energy* $\mathcal{E}_\xi$ to be a standard EBM as in (3) parameterized by $\xi$ mapping $\mathcal{X}^{d'} \to \mathbb{R}^+$. We then define the *conditional energy* $\mathcal{E}_\gamma$ to be an EBM representation of a conditional distribution on $\mathcal{X}^{d^*}$ given $x'$. $\mathcal{E}_\gamma$ maps $\mathcal{X}^{d^*} \times \mathcal{X}^{d'} \to \mathbb{R}^+$, which permits the following conditional density model:

$$p_\gamma(x^*|x') = \frac{1}{Z^*_{\gamma,x'}} \exp(-\mathcal{E}_\gamma(x^*|x')) \tag{4}$$

$$Z^*_{\gamma,x'} = \int_{x^*} \exp(-\mathcal{E}_\gamma(x^*|x'))$$

Since $\mathcal{E}_\gamma$ represents the energy for a conditional density with respect to $x'$, we apply the normalization over $x^*$ rather than $x$. We further define a *marginal generator* $\mathcal{G}_\omega$ mapping $\mathcal{Z}^l \to \mathcal{X}^{d'}$ and a *conditional generator* $\mathcal{G}_\upsilon$ mapping $\mathcal{Z}^{l'} \times \mathcal{X}^{d'} \to \mathcal{X}^{d^*}$. Samples for $x$ are generated using learnt PPEMs by first sampling $z$ and $z'$ from the latent distributions of the marginal and conditional generators and then generating $x' \leftarrow \mathcal{G}_\xi(z)$ followed by $x^* \leftarrow \mathcal{G}_\gamma(z'|x')$.

The motivating goals of the dual energy and generator framework of PPEMs are (i) it isolates the representation for $x'$ from $x^*$ (and its relation to $x'$), allowing a privacy penalty to be applied when learning the representation for $x'$ that does not affect the attributes which do not require $\alpha$-LMDP protection; (ii) it provides an explicit representation for $p_{\mathcal{G}_{x'}}$, which is required for constructing the goodness of fit test for $\alpha$-LMDP. This permits an adversarial training procedure that results in generative models which satisfy $\alpha$-LMDP for $x'$ without affecting $x^*$ and its relation to $x'$.

We next discuss this training procedure in a general, non-private setting and show a Nash equilibrium exists where the generators produce samples indistinguishable from the real data. We then show how $\alpha$-LMDP and DP protection can be incorporated into the procedure. We note PPEMs *do not* require estimating partition functions or expensive MCMC-based procedures to generate samples, as samples come directly from learnt generators. Pseudocode for training and sampling is provided in Appendix A and proofs are provided in Appendix B. Code is also provided in the attached supplement.

### 3.3 Training PPEMs

Our goal in training the two EBMs which comprise a PPEM is to learn $\mathcal{E}_\xi$ and $\mathcal{E}_\gamma$ which together represent the joint distribution of $x$ according to the following likelihood model:

$$\begin{aligned} \log p_\theta(x) &= \log \left( p_\gamma(x^*|x') \, p_\xi(x') \right) \\ &= -\mathcal{E}_\gamma(x^*|x') - \log Z^*_{\gamma,x'} - \mathcal{E}_\xi(x') - \log Z_\xi \end{aligned} \tag{5}$$

Notice that the gradients $\nabla_\gamma \log p_\theta$ and $\nabla_\xi \log p_\theta$ do not include terms from the marginal and conditional EBMs, respectively, so we can learn an EBM for the joint density $p_\theta$ by considering $\mathcal{E}_\xi$ and $\mathcal{E}_\gamma$ separately. We thus train $\mathcal{E}_\xi$ and $\mathcal{E}_\gamma$ adversarially with $\mathcal{G}_\omega$ and $\mathcal{G}_\upsilon$, respectively.

Our trained $\mathcal{E}_\xi$ should assign low energies to real samples and high energies to the samples generated by $\mathcal{G}_\omega$. As in Zhao et al. (2017), we use the following margin loss, where $m$ is a positive margin and $[\cdot]^+ = \max(0, \cdot)$, and use a separate generator loss (Goodfellow et al., 2014):

$$L_\xi\left(x', z\right) = \mathcal{E}_\xi\left(x'\right) + \left[m - \mathcal{E}_\xi\left(\mathcal{G}_\omega(z)\right)\right]^+ \tag{6}$$
$$L_\omega(z) = \mathcal{E}_\xi\left(\mathcal{G}_\omega(z)\right)$$

We train $\mathcal{E}_\xi$ and $\mathcal{G}_\omega$ using the above losses by iteratively taking minibatchs of training data $\{x'_i\}_{i=1}^n$ and samples from the generator's latent distribution and optimizing the EBM in an inner loop and generator in an outer loop. In practice, it is computationally prohibitive to optimize the EBM to convergence at each outer step so $k$ inner steps are performed for each outer step. Adversarial training of the conditional model is carried out identically, independent of $\mathcal{E}_\xi$ and $\mathcal{G}_\omega$, but while conditioning on real samples from $p_{x'}$. We use the following losses for $\mathcal{E}_\gamma$ and $\mathcal{G}_\upsilon$:

$$L_\gamma\left(x^*, z', x'\right) = \mathcal{E}_\gamma\left(x^*|x'\right) + \left[m - \mathcal{E}_\gamma\left(\mathcal{G}_\upsilon\left(z'|x'\right)|x'\right)\right]^+ \tag{7}$$
$$L_\upsilon\left(z', x'\right) = \mathcal{E}_\gamma\left(\mathcal{G}_\upsilon\left(z'|x'\right)|x'\right)$$

When training $\mathcal{E}_\gamma$ and $\mathcal{G}_\upsilon$, we sample a minibatch $\{x_i^*, x_i'\}_{i=1}^n$ and provide $\{x_i'\}_{i=1}^n$ to both $\mathcal{E}_\gamma$ and $\mathcal{G}_\upsilon$ to condition on. Otherwise, the procedure is the same as for the marginal model, substituting $x_i^*$ for $x_i'$ and $z_i'$ for $z_i$. We emphasize that when training $\mathcal{E}_\gamma$ and $\mathcal{G}_\upsilon$, we condition on values $x'$ from the *real data* for both. This ensures that $\mathcal{E}_\gamma$ learns to provide strong coverage as a conditional density model for $x^*$ when conditioning on $x' \sim p_{x'}$ and similarly $\mathcal{G}_\upsilon$ produces realistic samples. It is important that $\mathcal{G}_\upsilon$ be trained by conditioning on the real marginals $x' \sim p_{x'}$ rather than marginals generated by $\mathcal{G}_\omega : x' \sim p_\omega$, since $\mathcal{E}_\gamma$ might otherwise learn to assign high energies to samples generated by $\mathcal{G}_\upsilon$ because of differences between $p_{x'}$ and $p_\omega$ rather than the quality of samples produced by $\mathcal{G}_\upsilon$. This is important because if PPEMs are trained to ensure $\alpha$-LMDP for $x'$ then, by definition, such differences between the real and generative distributions for $x'$ will exist.

### 3.4 Optimality of the solution

We can show that when the systems defined in (6) and (7) converge, the generators produce samples indistinguishable from the real data. Define the following:

$$V_m\left(\xi, \omega\right) = \int_{x', z} L_\xi\left(x', z\right) p_{x'}\left(x'\right) p_z\left(z\right) \tag{8}$$

$$U_m\left(\xi, \omega\right) = \int_z L_\omega(z) p_z\left(z\right) \tag{9}$$

$$V_c\left(\gamma, \upsilon\right) = \int_{x, z'} L_\gamma\left(x^*, z', x'\right) p_x(x) p_{z'}\left(z'\right) \tag{10}$$

$$U_c\left(\gamma, \upsilon\right) = \int_{z', x'} L_\upsilon\left(z', x'\right) p_{z'}\left(z'\right) p_{x'}\left(x'\right) \tag{11}$$

A Nash equilibrium for the first system is a pair $(\xi^*, \omega^*)$ such that $\forall \xi$, $V_m(\xi^*, \omega^*) \leq V_m(\xi, \omega^*)$ and $\forall \omega$, $U_m(\xi^*, \omega^*) \leq U_m(\xi^*, \omega)$. A Nash equilibrium for the second system is a pair $(\gamma^*, \upsilon^*)$ such that $\forall \gamma$, $V_c(\gamma^*, \upsilon^*) \leq V_c(\gamma, \upsilon^*)$ and $\forall \upsilon$, $U_c(\gamma^*, \upsilon^*) \leq U_c(\gamma^*, \upsilon)$. The results below follow directly from Theorem 1 and Theorem 2 in Zhao et al. (2017) when accounting for the partitioning of $x$.

**Theorem 3.1.** *If $(\xi^*, \omega^*)$ and $(\gamma^*, \upsilon^*)$ are Nash equilibria for the systems (6) and (7), then $p_{\omega^*} p_{\upsilon^*} = p_x$ almost everywhere and $V_m(\xi^*, \omega^*) = V_c(\gamma^*, \upsilon^*) = m$.*

**Theorem 3.2.** *Nash equilibria for (6) and (7) exist and are characterized by (a) $p_{\omega^*} p_{\upsilon^*} = p_x$ almost everywhere and (b) there exists a constant $\nu_m \in [0, m]$ such that $\mathcal{E}_{\xi^*}(x') = \nu_m$ almost everywhere and for $x' \in \mathcal{X}^{d'}$ a constant $\nu_c \in [0, m]$ such that $\mathcal{E}_{\gamma^*}(x^*|x') = \nu_c$ almost everywhere.*

### 3.5 Incorporating $\alpha$-LMDP Protection into PPEM Training

When $x'$ corresponds to attributes requiring $\alpha$-LMDP protection, we only need to modify the training procedure for the marginal model to ensure the learnt generative distribution is different from $p_{x'}$, since $\mathcal{G}_v$ only generates attributes $x^*$ for a given setting of $x'$. Our goal is to augment training in such a way that the resulting generative distribution is sufficiently different from $p_{x'}$ but *individual* samples remain realistic.

For example, consider the simple univariate case where $p_{x'}$ is $\mathcal{N}(0,1)$. A generator that produces samples roughly uniform between -1 and 1 would likely satisfy $\alpha$-LMDP while still producing values for $x'$ that are individually realistic given a training dataset.

To achieve this goal, we retain the loss functions defined in (6), but append a soft penalty based on the goodness of fit test statistic we use to satisfy $\alpha$-LMDP. This ensures training promotes the dual objective that samples are *individually* realistic but *in the aggregate* satisfy $\alpha$-LMDP. Penalizing $\mathcal{G}_\omega$ directly, however, may result in $\mathcal{E}_\xi$ initially learning to trivially distinguish real from generated samples and $\mathcal{G}_\omega$ never receiving sufficient signal to learn to generate realistic samples. Instead, we penalize $\mathcal{E}_\xi$ and rely on the fact that since $\mathcal{G}_\omega$ interacts with the data only indirectly through $\mathcal{E}_\xi$, its representation power is limited by $\mathcal{E}_\xi$.

To implement this penalization for $\alpha$-LMDP protection, we require a goodness of fit test that is sufficiently flexible to make use of our EBM-based representation for the marginal distribution $p_\xi$. In addition, we desire a test that does not require evaluating the partition function $Z_\xi$ and can be calculated efficiently for each minibatch $\{x_i'\}_{i=1}^n$.

The *Kernel Stein Discrepancy (KSD)* (Liu et al., 2016) is a powerful nonparametric goodness of fit test which has these properties. The KSD only interacts with a density $p$ through its *score function* $\mathfrak{s}_p(x) = \nabla_x \log p(x)$. When $p$ has an EBM-based representation, $\mathfrak{s}_p$ does not depend on the partition function:

$$\mathfrak{s}_p(x) = \nabla_x \log\left[Z_\theta^{-1} \exp\left(-\mathcal{E}_\theta(x)\right)\right] = -\nabla_x \mathcal{E}_\theta(x)$$

Thus, computing the KSD for $p_\xi$ does not require evaluating $Z_\xi$.

The KSD is based on *Stein's identity*, which states that for distributions $p$ and $q$ with support on $\mathbb{R}$ and smooth functions $f$, the following equivalence holds if and only if $p = q$ (Stein et al., 2004):

$$\mathbb{E}_p\left[\mathfrak{s}_q(x)f(x) + \nabla_x f(x)\right] = 0 \tag{12}$$

A *Stein discrepancy* $\mathbb{S}(p,q)$ between $p$ and $q$ can thus be defined by taking a maximum over $f$ for a sufficiently rich space of functions $\mathcal{F}$:

$$\mathbb{S}(p,q) = \max_{f \in \mathcal{F}} \ \mathbb{E}_p\left[\mathfrak{s}_q(x)f(x) + \nabla_x f(x)\right]^2 \tag{13}$$

When $\mathcal{F}$ is a reproducing kernel Hilbert space (RKHS) associated with a smooth positive definite kernel $k(\cdot,\cdot)$, e.g. the RBF function $\exp(-\sigma^{-1}\|x - x'\|_2)$, $\mathbb{S}(p,q)$ is the Kernel Stein Discrepancy (KSD).

Liu et al. (2016) provides the following empirical estimator for the KSD:

$$\hat{\mathbb{S}}(p,q) = \frac{1}{n(n-1)} \sum_{i \neq j} u_q(x_i, x_j) \tag{14}$$

$$\begin{aligned} u_q(x,x') = {}& \mathfrak{s}_q(x)^\top k(x,x')\mathfrak{s}_q(x') + \nabla_{x'}\mathfrak{s}_q(x)^\top k(x,x') \\ & + \nabla_x k(x,x')^\top \mathfrak{s}_q(x') + tr(\nabla_{x,x'} k(x,x')). \end{aligned} \tag{15}$$

The computational cost of evaluating $\hat{\mathbb{S}}(p,q)$ at each gradient update is $\mathcal{O}(n^2)$ for a minibatch of size $n$. Thus Liu et al. (2016) also proposes the following more efficient estimator:

$$\hat{\mathbb{S}}_l(p,q) = \frac{1}{\lfloor n/2 \rfloor} \sum_{i=1}^{\lfloor n/2 \rfloor} u_q(x_{2i-1}, x_{2i}) \tag{16}$$

$\hat{\mathbb{S}}_l(p,q)$ has $\mathcal{O}(n)$ computational complexity at a cost of less statistical power (Liu et al., 2016). There are two issues, however, with including $\hat{\mathbb{S}}_l(p,q)$ in a soft penalty that can lead to training instability: (i) its

magnitude can vary significantly across minibatches; (ii) oscillations between positive and negative values can occur. To address these issues, we propose the following non-negative, normalized variant of the KSD, which we show has a $\chi_1^2$ null distribution:

$$\hat{\mathbb{S}}_{l,n}(p,q) = \frac{(\lfloor n/2 \rfloor - 1) \left[ \sum_{i=1}^{\lfloor n/2 \rfloor} u_q(x_{2i-1}, x_{2i}) \right]^2}{\lfloor n/2 \rfloor \sum_{i=1}^{\lfloor n/2 \rfloor} u_q(x_{2i-1}, x_{2i})^2} \tag{17}$$

**Lemma 3.1.** *$\hat{\mathbb{S}}_{l,n}(p,q)$ has an asymptotic $\chi_1^2$ distribution under the null hypothesis $p = q$.*

$\hat{\mathbb{S}}_{l,n}(p,q)$ is more well behaved than $\hat{\mathbb{S}}_l(p,q)$ when used in a soft penalty, but has the same computational efficiency and performance as a test statistic (in terms of type I and II errors), as we confirm empirically in section 5. We propose the following soft penalty based on $\hat{\mathbb{S}}_{l,n}(p,q)$ which grows in magnitude relative to the confidence with which the hypothesis $H_0 : \{x_i'\}_{i=1}^n \sim p_\xi$ may be rejected:

$$L_\xi^{pen}(x', z) = L_\xi(x', z) + \lambda_\alpha \left[ F_{\chi_1^2}^{-1}(1-\alpha) - \hat{\mathbb{S}}_{l,n}(p_{x'}, p_\xi) \right]^+ \tag{18}$$

$F_{\chi_1^2}^{-1}$ is the $\chi_1^2$ quantile function and $\lambda_\alpha$ is a hyperparameter used to scale the relative magnitudes of the two quantities. Since $L_\xi$ and $-\hat{\mathbb{S}}_{l,n}(p_{x'}, p_\xi)$ have conflicting goals, the influence of $-\hat{\mathbb{S}}_{l,n}(p_{x'}, p_\xi)$ is modulated by the critical value of the level$-\alpha$ test: it is inactive whenever $H_0$ can be rejected with confidence level $1 - \alpha$ and otherwise decreases relative to the confidence at which $H_0$ may be rejected. We can now state the following result which relates training using the penalized loss (18) to satisfying $\alpha$-LMDP:

**Theorem 3.3.** *For sufficiently large $\lambda_\alpha$ and $\mathcal{G}\omega$ and $\mathcal{E}_\xi$ with sufficient representation power, when $\mathcal{E}_\xi$ is trained to convergence using (18) and reaches a global optimum, the resulting $\mathcal{G} = (\mathcal{G}_\omega, \mathcal{G}_\upsilon)$ will satisfy $\alpha$-LMDP with respect to $x'$ and the KSD test using the estimator $\hat{\mathbb{S}}_{l,n}(p_{x'}, p_\xi)$.*

The above result assumes $\mathcal{E}_\xi$ is trained to convergence, which may be computationally prohibitive, but this is not a practical limitation since after training we can confirm the model satisfies $\alpha$-LMDP using the KSD and adjust the hyperparameters and retrain if needed.

### 3.6   Incorporating Differential Privacy into PPEM Training

DP can also be incorporated into PPEMs by perturbing gradients when training the marginal and conditional EBMs, as in DP-GAN. Following Abadi et al. (2016), we clip gradients to ensure their norms are bounded by a constant $C$ and then add Gaussian noise, scaled according to $C$ and a factor $\sigma$ determining the $(\epsilon, \delta)$ level of privacy. We use the moments accountant method of Abadi et al. (2016) to find a setting of $\sigma$ which will result in the desired level of privacy given the sample size, minibatch size and number of iterations. The post-processing property of DP (Dwork & Roth, 2014) guarantees the generators are $(\epsilon, \delta)$-DP with respect to their generated attributes since each observe these only through their corresponding EBM.

### 3.7   Stabilizing Training

While existing work demonstrates that EBM-based generative models for images can be successfully learned using adversarial training, pairing an EBM with a generator (Zhao et al., 2017; Wang & Liu, 2017), our experiments using tabular datasets found that gradients become very unstable and regularization is needed for such data. While image data is superficially more complex than tabular data, it is highly structured due to spacial similarities and distributions of pixel values, whereas tabular data may be non-Gaussian, multimodal and highly imbalanced. Other generative modeling approaches for tabular data note similar difficulties (Xu et al., 2019). Prior work interprets imposing Lipschitz constraints on EBMs as a form of smoothing regularization (Che et al., 2020). Given the success of using the soft penalty defined in Gulrajani et al. (2017) to enforce Lipschitz constraints on Wasserstein GAN critics and similarities between the critic and EBM-based loss functions, we incorporate the same Lipschitz soft penalty into (6) and (7) when optimizing the marginal and conditional EBMs.

### 3.8 Network architectures

We use multi-layer perceptron (MLP) architectures with 3-layers and LeakyReLU activations for EBMs and generators. Despite findings that autoencoder-based EBM architectures perform well with image data (Wang & Liu, 2017; Zhao et al., 2017), our experiments indicated MLP architectures perform best with tabular datasets. We add residual connections and use batch normalization in the generators. For the conditional EBM and generator, we include initial sets of neurons connected only to the conditional and unconditional attributes. These architectures are depicted in figure 1 (right).

## 4 Related Work

### 4.1 Privacy of data distributions

Zhang et al. (2022) proposes two notions of *attribute privacy* for a dataset or distribution: the first notion, which is the closest to $\alpha$-LMDP, considers protecting specific statistical properties of a dataset, whereas the second notion considers protecting parameters of a distribution with dependencies that can be formalized in terms of a Bayesian network. The most important differences between the first definition and $\alpha$-LMDP are (i) the first definition requires choosing specific functions of columns of a dataset to be protected whereas $\alpha$-LMDP considers general differences between marginal distributions that can be detected using a two-sample test, (ii) the first definition requires a multivariate Gaussian assumption (or close to multivariate Gaussian) for the data and the functions of the columns of the dataset must be linear, whereas $\alpha$-LMDP is fully nonparametric and (iii) the first definition is mechanism-based, whereas $\alpha$-LMDP is model-based. We discuss considerations for using each definition in section 6.2. Suri & Evans (2022) proposes a general definition for *distribution inference attacks* and considers a class of attacks for distinguishing between different possible training distributions of a classifier. Balle et al. (2020) considers a hypothesis testing interpretation of differential privacy; this interpretation considers testing whether a specific individual contributed their data rather than global properties of a dataset.

### 4.2 Generative models for healthcare and finance

Procedures leveraging domain knowledge have been proposed in healthcare (Buczak et al., 2010; Park et al., 2013) and finance (Galbiati & Soramäki, 2011; Weber et al., 2019), but there are few domain-agnostic approaches. medGAN (Choi et al., 2017) is one such procedure for binary and count variables. medGAN uses an autoencoder that is pretrained on the real data; the GAN generator learns to generate from the autoencoder latent dimension while the discriminator compares the real and fake outputs of the autoencoder's decoder. GAN architectures for financial time series have also been proposed (Wiese et al., 2020). CT-GAN is another GAN architecture for tabular data with mixed discrete and continuous variables (Xu et al., 2019).

### 4.3 Generative models with privacy-preserving properties

DP-GAN (Xie et al., 2018) and PATE-GAN (Yoon et al., 2019) are GANs which both incorporate DP. DP-GAN uses DP-SGD whereas PATE-GAN is based on the PATE framework (Papernot et al., 2018). PATE-GAN is evaluated using both financial and healthcare data.

### 4.4 EBM-based generative models

EnergyGAN (Zhao et al., 2017) and SteinGAN (Wang & Liu, 2017) are related approaches which combine EBMs with generators and use adversarial training to generate synthetic images. EnergyGAN's training is identical to our marginal model training when no privacy penalty or gradient stabilization is used. SteinGAN uses an alternative procedure, *Stein variational gradient descent*, to train the energy model. Both use AE architectures for the energy model whereas we use MLPs.

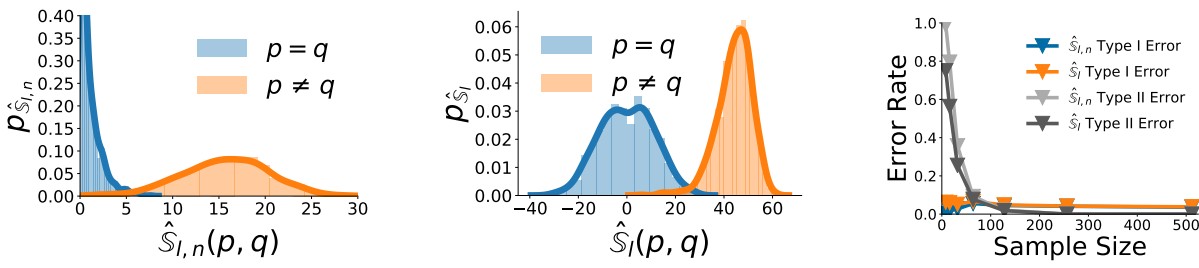

Figure 2: Distributions of $\hat{\mathbb{S}}_{l,n}$ (left) and $\hat{\mathbb{S}}_l$ (center) empirical type I and II error rates (right)

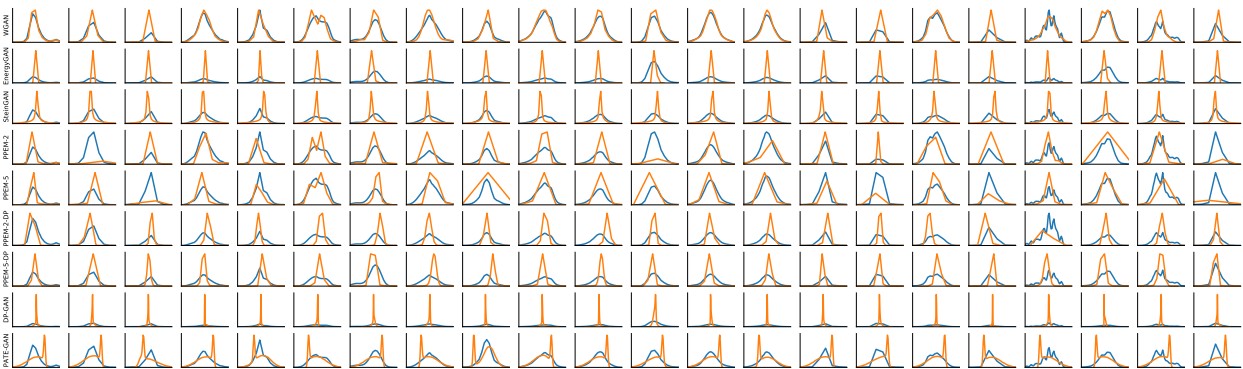

Figure 3: KDE plots for real (blue) and synthetic data (orange) for attributes not requiring $\alpha$-LMDP protection

## 5 Experiments

In all experiments, we set $\alpha = .05$ for $\alpha$-LMDP and $\epsilon = 1$, $\delta = n^{-1}$ for DP. We use RBF kernels and choose bandwidths using the median distance heuristic described in (Garreau et al., 2018) when evaluating the KSD. Further details regarding datasets, hyperparameters and additional results are provided in Appendices C and D.

### 5.1 Evaluation of KSD variant using synthetic data

We first confirm our proposed KSD variant $\hat{\mathbb{S}}_{l,n}$ has the desired properties from section 3 and the same test performance as $\hat{\mathbb{S}}_l$ using simulated data from a 1-D Gaussian mixture model (GMM) with 5 equally weighted components. Since the underlying distribution is known, we can calculate the score function exactly when evaluating the KSD. We evaluate $\hat{\mathbb{S}}_{l,n}$ and $\hat{\mathbb{S}}_l$ across 10,000 sets of 128 samples, the minibatch size used to train PPEMs, using data sampled from the true GMM ($p = q$) and an alternative 5-component GMM with different parameters ($p \neq q$).

Figure 2 shows the empirical distributions of each estimator under the null and alternative conditions. We observe $\hat{\mathbb{S}}_{l,n}$ is non-negative and falls within the expected range of a $\chi_1^2$ variable under the null condition, whereas $\hat{\mathbb{S}}_l$ is negative for some samples and has significant variance, confirming $\hat{\mathbb{S}}_{l,n}$ has the properties from section 3 which result in a more well behaved soft penalty. To compare test performance, we report empirical type I and II errors when simulating from GMMs with randomly chosen parameters 10,000 times in figure 2. We observe type I errors are well controlled for both $\hat{\mathbb{S}}_{l,n}$ and $\hat{\mathbb{S}}_l$ and the two estimators exhibit comparable statistical power, confirming $\hat{\mathbb{S}}_{l,n}$ has the same test performance as $\hat{\mathbb{S}}_l$.

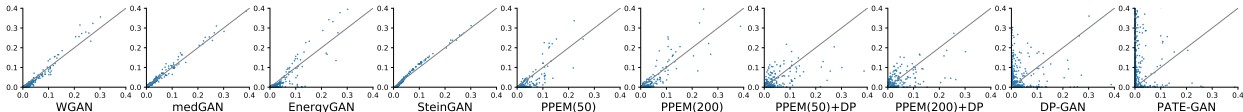

Figure 4: MIMIC-III Bernoulli success probability for the real (x-axis) vs. generated (y-axis) data

Table 1: KSD and MMD test results and classification and regression performance

| | Kaggle Credit Card Fraud | | | | | MIMIC-III EHR | |
|---|---|---|---|---|---|---|---|
| | $p_{KSD}$ | $p_{MMD}$ | AUROC | MSE | $R^2$ | $p_{KSD}$ | AUROC |
| REAL DATA | - | - | 0.975 | 0.124 | 0.881 | - | 0.970 |
| WGAN (NO PRIVACY) | - | 0.15 | 0.966 | 0.126 | 0.879 | - | 0.811 |
| MEDGAN (NO PRIVACY) | - | - | - | - | - | - | 0.625 |
| ENERGYGAN (NO PRIVACY) | - | 0.00 | 0.871 | 7.13 | -5.86 | - | 0.594 |
| STEINGAN (NO PRIVACY) | - | 0.00 | 0.822 | 1.81 | -0.740 | - | 0.508 |
| PPEM (2/50) | 0.00 | 0.02 | 0.922 | 0.184 | 0.822 | 0.00 | 0.828 |
| PPEM (5/200) | 0.00 | 0.00 | 0.948 | 0.196 | 0.812 | 0.00 | 0.769 |
| PPEM (2/50) + DP | 0.00 | 0.00 | 0.872 | 0.656 | 0.368 | 0.00 | 0.504 |
| PPEM (5/200) + DP | 0.00 | 0.00 | 0.889 | 0.485 | 0.533 | 0.00 | 0.511 |
| DP-GAN | - | 0.00 | 0.857 | 1.881 | -0.811 | - | 0.491 |
| PATE-GAN | - | 0.00 | 0.879 | 1.569 | -0.510 | - | 0.490 |

## 5.2 Evaluation of PPEMs using real data

We next apply PPEMs to real financial and healthcare datasets that have previously been used to benchmark privacy-preserving generative models: the Kaggle credit card fraud dataset (Pozzolo et al., 2015), used as the primary evaluation dataset for PATE-GAN, consists of 28 factors used to predict whether a transaction is fraudulent and the transaction amount; the MIMIC-III critical care electronic healthcare record (EHR) dataset (Johnson et al., 2016) consists of binary indicators for diagnoses patients received. For MIMIC-III, we use the same procedure as Choi et al. (2017) and Xie et al. (2018) to construct 1070 longitudinal indicators for each patient and then add extremely-low magnitude Gaussian noise so the binary indicators appear continuous, which we found led to improved performance over using the medGAN autoencoder architecture.

For each dataset we train PPEMs with $\alpha$-LMDP for two cases: when $x'$ consists of (i) 2 or 50 and (ii) 5 or 200 attributes for the Kaggle and MIMIC-III datasets, respectively, since MIMIC-III is a much higher dimensional dataset. For each case, we train PPEMs with (a) $\alpha$-LMDP only and (b) $\alpha$-LMDP and DP (for both the marginal and conditional models). We evaluate the results to confirm $\alpha$-LMDP is satisfied and compare the fidelity of the generated samples to non-private (Wasserstein GAN, EnergyGAN, SteinGAN) and DP (DP-GAN, PATE-GAN) baselines. We also include medGAN for MIMIC-III.

### 5.2.1 Privacy evaluation

To confirm the trained PPEMs satisfy .05-LMDP, we evaluate the KSD with the learnt marginal EBMs and training data for the protected attributes ($x'$) and calculate a $p$-value, corresponding to the minimum level-$\alpha$ test with which we can reject the null hypothesis that the training data for $x'$ follow the learnt energy-based distribution. We also evaluate the MMD two sample test (Gretton et al., 2012) for whether the real and generated data for the protected attributes ($x'$) follow the same distribution and calculate a $p$-value in the case of the real-valued credit card transaction data. We report these in table 1. We observe that the $p$-values for PPEMs under all conditions (for the KSD and MMD and each dataset) are $\leq .05$, confirming .05-LMDP is satisfied. We note that achieving a $p$-value $\leq 0.05$ is non-trivial as the p-value for WGAN is 0.15; the small $p$-values for EnergyGAN and SteinGAN are due to poor fidelity, which is discussed in the following section.

### 5.2.2 Fidelity evaluation of credit card transaction data

We next consider the fidelity of the credit card transaction data, following previous benchmarking approaches. In figure 3, we plot kernel density estimates for the attributes which do not require $\alpha$-LMDP protection ($x^*$) in the real (blue) vs. generated (orange) data for each PPEM training scenario mentioned above and baseline. We also train logistic regression models to predict whether a transaction is fraudulent and linear regression models to predict the transaction amount using the generated data for training and evaluate these models' performance on the real data in terms of area under the ROC curve (AUROC) and mean-squared error (MSE) and coefficient of determination ($R^2$), respectively. We report these in table 1 and include an additional baseline where the prediction models are trained using the real data ("Real Data").

We first observe that the non-private Wasserstein GAN produces synthetic data with marginal distributions which best match the real data, whereas EnergyGAN and SteinGAN result in more peaked distributions which fail to capture the true marginal distributions. The poor performance of EnergyGAN and SteinGAN is due to unstable gradients when using tabular data and consistent with our expectations discussed in section 3: *Stabilizing Training*, which motivates the gradient stabilization strategy used for PPEMs. We note that the performance of EnergyGAN and SteinGAN in some cases is worse than methods which include privacy due to poor fidelity resulting from gradient stability issues. We include the EnergyGAN and SteinGAN results for completeness, relying on the Wasserstein GAN results as the primary non-private benchmark.

We next observe that the plots for PPEMs trained without DP are the next closest to the real data (after the Wasserstein GAN), indicating the privacy vs. accuracy tradeoff when learning models which satisfy $\alpha$-LMDP is (at least) no worse than the tradeoff for $(1, n^{-1})-$DP. The plots for the PPEMs with $\alpha$-LMDP and DP are comparable to the plots for PATE-GAN (DP-GAN is slightly worse), indicating that when .05-LMDP is combined with $(1, n^{-1})-$DP, it does not significantly affect the privacy vs. accuracy tradeoff. We observe the same pattern when we consider the prediction models trained on the synthetic data and evaluated on the real data: the performance for the PPEMs without DP is greater than all DP approaches and the performance for the PPEMs with $\alpha$-LMDP and DP is comparable to PATE-GAN.

### 5.2.3 Fidelity evaluation of electronic healthcare records

Finally, we consider the fidelity of the EHR data, again following previous benchmarking approaches. We report average AUROC when the generated data is used to train classifiers for each attribute, using the other attributes as predictors, and evaluation is carried out using the real data in table 1. We also plot Bernoulli success probabilities for each attribute in the real vs. generated data in figure 4 as in Choi et al. (2017): ideally, it should form a diagonal line.

Again, we observe the Wasserstein GAN is the best non-private baseline (SteinGAN closely matches the data in terms of Bernoulli success probabilities, but fails to capture relationships between variables as evidenced by low average AUROC, again due to unstable gradients), the PPEMs trained with $\alpha$-LMDP but without DP most closely resemble the real data and WGAN baseline compared to other privacy-preserving approaches, and the PPEMs trained with $\alpha$-LMDP and DP perform comparably to PATE-GAN.

### 5.3 Summary of results

In summary, from these results we draw the following conclusions: (i) PPEMs succeed in ensuring $\alpha$-LMDP is satisfied; (ii) the fidelity of data generated using PPEMs compared to that of DP generative models reveals the privacy vs. accuracy tradeoff for $\alpha$-LMDP, when implemented using PPEMs, is (at least) no worse than the tradeoff for DP; (iii) when PPEMs are trained to ensure both $\alpha$-LMDP and DP, the results are comparable to using only DP; (iv) the non-negative normalized variant of the KSD we propose $\hat{\mathbb{S}}_{l,n}$ results in a more stable soft penalty, but retains the same test performance of $\hat{\mathbb{S}}_l$; (v) the gradient stabilization strategy of PPEMs successfully prevents the instabilities which result when applying EnergyGAN and SteinGAN to tabular data.

# 6 Discussion

We proposed $\alpha$-LMDP, a new privacy notion, and PPEMs, novel energy-based generative models which support training with $\alpha$-LMDP as well as DP, if required. Our results confirm that PPEMs succeed in ensuring $\alpha$-LMDP is satisfied while maintaining a favorable privacy vs. fidelity tradeoff. Below, we discuss limitations of $\alpha$-LMDP and PPEMs as well as potential broader impacts.

## 6.1 Limitations and Future Work

While $\alpha$-LMDP provides a flexible framework for cases where a data owner wishes to create general differences between an observed marginal distribution and the resulting generative distribution and achieve plausible deniability among these variables, it does *not*, however, guarantee that any specific individual function of these variables is different in the generative distribution, e.g. the mean of a specific variable. This is an important distinction between $\alpha$-LMDP and the first proposed notion of attribute privacy in Zhang et al. (2022); our approach provides flexible protection for a general distribution without parametric assumptions and requiring that specific functions are choosen to protect, whereas the first privacy notion in Zhang et al. (2022) provides stronger protection for specific functions that are chosen, but with limitations on the types of functions that may be used and parametric assumptions for the distribution. A potential area of future research might consider combining these approaches in a generative model to offer flexible general protection for marginal distributions with stronger protection for specific sensitive properties.

There are several limitations of the PPEM architecture and training paradigm. In terms of the protection offered, PPEMs are limited in that they only offer protection for a single marginal distribution, whereas in many cases there may be multiple disjoint sets of variables for which marginal distribution protection is required. The proposed setup requires these sets of variables to be combined into a single marginal distribution, which may be suboptimal in terms of generative fidelity, and there is no straightforward way to extend the PPEM model to enable protection for multiple disjoint variable sets without making additional assumptions. PPEMs also assume a GAN style adversarial training procedure with energy models, whereas *diffusion models* (Sohl-Dickstein et al., 2015) have gained prominence more recently, often outperforming GANs on tasks like image generation (Dhariwal & Nichol, 2021). It is not straightforward, however, how the KSD might be integrated into the diffusion model training procedure efficiently; the difficulties associated without incorporating the KSD into a generative model training procedure motivated our choice of energy models with adversarial training. Future work might address these limitations.

A more practical concern for the PPEM model is that when generating data, one might encounter generated $x'$ which does not map to realistic outputs for $x^*$ in the conditional model, since the conditional model is trained using only real data for $x'$. While we did not encounter issues resulting from this in our experiments, the addition of a sampling layer to the conditional architecture, as in variational autoencoders (Kingma & Welling, 2014), might improve robustness.

Finally, future work might also consider other goodness of fit tests, such as the Finite Set Stein Discrepancy (FSSD) (Jitkrittum et al., 2017), which is a more powerful test than the linear KSD, but difficult to incorporate into a soft penalty that scales with the rejection level due to its null distribution.

## 6.2 Broader Impacts

As we emphasize throughout the paper, while $\alpha$-LMDP protects sensitive aggregate level information about a data set, it *does not* protect the privacy of individual data points, which may be collected from individuals, as DP does. The misapplication of $\alpha$-LMDP with the expectation of the protection offered by DP could result in significant violations of the privacy of individuals. It is incumbent upon data owners and model builders to ensure the correct notion of privacy is used to achieve their desired privacy objectives.

While the first notion of attribute privacy introduced in Zhang et al. (2022) is closer to $\alpha$-LMDP than DP, these two notions should also not be used interchangeably. As we note in section 6.1, $\alpha$-LMDP provides flexible general protection for sensitive marginal distributions without specifying explicit functions to protect and making parametric assumptions, but does not guarantee any explicit function of the marginal distribution

will be different from the data distribution in the generative model. This makes $\alpha$-LMDP an optimal choice for cases where a data owner wishes to create plausible deniability across a range of properties of the marginal distribution without assuming an explicit parametric model for the data distribution, but does require explicit differences across any specific functions, e.g. a mean. The use of $\alpha$-LMDP in cases where a specific function must be different in the generative distribution may result in violations of the privacy of this function. Similarly, the use of the first notion of attribute privacy introduced in Zhang et al. (2022) in cases where protection is required for additional properties of the marginal distribution beyond linear functions or in cases where a multivariate Gaussian assumption is not appropriate may result in violations of aggregate dataset privacy expectations. Data owners must consider the competitive risks associated with leakage of different types of aggregate level information and choose the appropriate privacy definition (or choose not to proceed if neither suffice). While the potential impacts on data owners from misapplying these two notions of privacy have a distinctly different risk profile than failing to use DP when appropriate, it remains a significant risk when competitive information is discernible from aggregate properties of a dataset.

While training generative models with $\alpha$-LMDP has similarities to *fair generative modeling* (Choi et al., 2020), which seeks to learn generative models with fairness properties, it is important to recognize that PPEMs (and other hypothetical models trained with $\alpha$-LMDP) do not guarantee fairness properties, regardless of whether protected variables like race or gender are included in the specified marginal distribution. The goal of fair generative modelling is often to ensure the resulting generative distribution is as close as possible to a target (unbiased) distribution; $\alpha$-LMDP, however, seeks only to ensure the generative distribution is different from the observed data distribution, regardless of whether the resulting distribution is fair or unbiased with respect to any target distribution. Hence, including protected attributes in the marginal distribution for $\alpha$-LMDP not only does not guarantee fairness or unbiasedness for these attributes, it could actually result in a more biased generative distribution. It is thus incumbent upon model builders to be especially careful whenever protected attributes are included in marginal distributions when $\alpha$-LMDP is applied and consider the implications of any biases that may be replicated or introduced by PPEMs both in general and in potential downstream applications. Fairness must always be always be evaluated separately from marginal distribution privacy.

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

## A   Pseudocode for Training and Sampling

---

**Algorithm 1:** Training Privacy-Preserving Energy Models

---

**for** *iterations* 1 *to* $T_{\mathcal{G}}$ **do**

    **for** *iterations* 1 *to* $T_{\mathcal{E}}$ **do**

        Sample minibatch of real data $\{x_i', x_i^*\}_{i=1}^n$

        Sample $\{z_i\}_{i=1}^n \sim \mathcal{N}(\mathbf{0}_l, I_l)$ and $\{z_i'\}_{i=1}^n \sim \mathcal{N}(\mathbf{0}_{l'}, I_{l'})$

        Sample $\{u_i\}_{i=1}^n \sim Unif(0,1)$

        $\{L_\xi(x_i', z_i)\}_{i=1}^n \leftarrow \left\{ \mathcal{E}_\xi(x_i') + [m - \mathcal{E}_\xi(\mathcal{G}_\omega(z_i))]^+ \right\}_{i=1}^n$

        $\{\hat{x}_i'\}_{i=1}^n \leftarrow \{u_i x_i' + (1 - u_i)\mathcal{G}_\omega(z_i)\}_{i=1}^n$

        $\left\{ P_\xi^L(x_i', z_i) \right\}_{i=1}^n \leftarrow \left\{ \lambda_L \left( \left\| \nabla_{\hat{x}_i'} \mathcal{E}_\xi(\hat{x}_i') \right\|_2 - 1 \right)^2 \right\}_{i=1}^n$

        $P_\xi^\alpha \leftarrow \lambda_\alpha \left[ F_{\chi_1^2}^{-1}(1 - \alpha) - \hat{\mathbb{S}}_{l,n}(p_{x'}, p_\xi) \right]^+$

        $\{g_i'\}_{i=1}^n \leftarrow \left\{ \nabla_\xi \left( L_\xi(x_i', z_i) + P_\xi^L(x_i', z_i) + \frac{1}{n} P_\xi^\alpha \right) \right\}_{i=1}^n$

        $\{L_\gamma(x_i^*, z_i, x_i')\}_{i=1}^n \leftarrow \left\{ \mathcal{E}_\gamma(x_i^* | x_i') + [m - \mathcal{E}_\gamma(\mathcal{G}_\upsilon(z_i' | x_i') | x_i')]^+ \right\}_{i=1}^n$

        $\{\hat{x}_i^*\}_{i=1}^n \leftarrow \{u_i x_i^* + (1 - u_i)\mathcal{G}_\upsilon(z_i' | x_i')\}_{i=1}^n$

        $\left\{ P_\gamma^L(x_i^*, z_i', x_i') \right\}_{i=1}^n \leftarrow \left\{ \lambda_L \left( \left\| \nabla_{\hat{x}_i^*} \mathcal{E}_\gamma(\hat{x}_i^*) \right\|_2 - 1 \right)^2 \right\}_{i=1}^n$

        $\{g_i^*\}_{i=1}^n \leftarrow \left\{ \nabla_\gamma \left( L_\gamma(x_i^*, z_i', x_i') + P_\gamma^L(x_i^*, z_i', x_i') \right) \right\}_{i=1}^n$

        **if** *differentially private* **then**

            Sample $w \sim \mathcal{N}(\mathbf{0}_{|\xi|}, C^2 \sigma^2 I_{|\xi|})$

            $\{g_i'\}_{i=1}^n \leftarrow \left\{ g_i' \, / \, \max\left(1, \frac{\|g_i'\|_2}{C}\right) + \frac{w}{n} \right\}_{i=1}^n$

            Sample $v \sim \mathcal{N}(\mathbf{0}_{|\gamma|}, C^2 \sigma^2 I_{|\gamma|})$

            $\{g_i^*\}_{i=1}^n \leftarrow \left\{ g_i^* \, / \, \max\left(1, \frac{\|g_i^*\|_2}{C}\right) + \frac{v}{n} \right\}_{i=1}^n$

        $\xi \leftarrow \text{ADAM}\left(\frac{1}{n} \sum_{i=1}^n g_i', \xi\right)$

        $\gamma \leftarrow \text{ADAM}\left(\frac{1}{n} \sum_{i=1}^n g_i^*, \gamma\right)$

    $\omega \leftarrow \text{ADAM}\left(\frac{1}{n} \sum_{i=1}^n \nabla_\omega \mathcal{E}_\xi(\mathcal{G}_\omega(z_i)), \omega\right)$

    $\upsilon \leftarrow \text{ADAM}\left(\frac{1}{n} \sum_{i=1}^n \nabla_\upsilon \mathcal{E}_\gamma(\mathcal{G}_\upsilon(z_i' | x_i') | x_i'), \upsilon\right)$

---

---

**Algorithm 2:** Sampling from Privacy-Preserving Energy Models

---

Sample $\{z\}_{i=1}^n \sim \mathcal{N}(\mathbf{0}_d, I_d)$ and $\{z'\}_{i=1}^n \sim \mathcal{N}(\mathbf{0}_{d'}, I_{d'})$

$\{x_i'\}_{i=1}^n \leftarrow \{\mathcal{G}_\omega(z_i)\}_{i=1}^n$

$\{\tilde{x}_i^*\}_{i=1}^n \leftarrow \{\mathcal{G}_\upsilon(z_i' | x_i')\}_{i=1}^n$

$\{x_i\}_{i=1}^n \leftarrow \{x_i', x_i^*\}_{i=1}^n$

---

## B   Proofs of Theorems

### B.1   Proof of Theorem 2.1

*Proof.*

Let $\mathcal{X} = [B, B']$ be a bounded interval on $\mathbb{R}$, $X$ a random variable on $\mathcal{X}^d$ with CDF $F_X(x) = \prod_{j=1}^d \frac{x^j - B}{B' - B}$, i.e. the dimensions are independent $Uniform(B, B')$ variables and $\mathcal{D}_x = \{x_i\}_{i=1}^n$ a dataset sampled according to $F_X$.

Let $\mathcal{M}$ be a mechanism which always returns a generative model $\mathcal{G}$ which is a collection of $d$ independent random number generators over $\mathcal{X}$ which generate values for each dimension of $X$.

Now define $\mathcal{D}_y = \{y\}_{i=1}^n$ to be another dataset sampled according to $F_X$ such that $\mathcal{D}_x$ and $\mathcal{D}_y$ differ by only one datapoint and define $\mathcal{S} \subseteq \text{Range}(\mathcal{M})$.

Since $\mathcal{M}$ always returns a collection of random number generator over $\mathcal{X}$ for each dimension of $x$, the following equivalence holds trivially for any such $\mathcal{D}_x$ and $\mathcal{D}_y$:

$$\mathbb{P}\left[\mathcal{G}(\mathcal{D}_x) \in \mathcal{S}\right] = \mathbb{P}\left[\mathcal{G}(\mathcal{D}_y) \in \mathcal{S}\right]$$

It follows that for $\epsilon, \delta \in \mathbb{R}^+$, the following inequality holds:

$$\mathbb{P}\left[\mathcal{G}(\mathcal{D}_x) \in \mathcal{S}\right] \leq \exp(\epsilon)\mathbb{P}\left[\mathcal{G}(\mathcal{D}_y) \in \mathcal{S}\right] + \delta$$

Thus $\mathcal{M}$ is a $(\epsilon, \delta)$-differentially private for any $\epsilon, \delta \in \mathbb{R}^+$.

However, since $\mathcal{G}$ generates values uniformly between $B$ and $B'$ for each dimension of $x$, $\{x'\}_{i=1}^n \sim p_{\mathcal{G}_{x'}}$ for any partitioning $x = \{x', x^*\}$. Thus, for $\alpha < 1$, a level-$\alpha$ goodness of fit test will fail to reject $H_0 : \{x'_i\}_{i=1}^n \sim p_{\mathcal{G}_{x'}}$ so $\mathcal{M}$ is not guaranteed to return generative models which satisfy $\alpha$-LMDP for any $\alpha < 1$. $\square$

## B.2   Proof of Theorem 2.2

*Proof.*

Let $\mathcal{D}_x = \{x_i\}_{i=1}^n$ and $\mathcal{D}_y = \{y_i\}_{i=1}^n$ be two datasets which differ by only a single datapoint and let 1 be the index of the datapoint for which these two datasets differ.

Let $\mathcal{M}$ be a mechanism which always returns a generative model $\mathcal{G}$ that is a biased identity function, i.e. $\mathcal{G}$ reproduces elements from the input dataset $\{x_i\}_{i=1}^n$ non-uniformly, with the specific bias that $\mathcal{G}$ returns $x_1$ with probability 1 and all other elements with probability 0.

Thus, for any partitioning of the dimensions of $x$, $x = \{x', x^*\}$, $p_{\mathcal{G}_{x'}}$ is a single point mass distribution so any level-$\alpha$ goodness of fit test with sufficient power will reject $H_0 : \{x'_i\}_{i=1}^n \sim p_{\mathcal{G}_{x'}}$ unless the underlying distribution from which $\{x'_i\}_{i=1}^n$ is sampled is a single point mass. Thus, $\mathcal{M}$ is guaranteed to return a generative model $\mathcal{G}$ which satisfies $\alpha$-LMDP whenever the underlying distribution is not a single point mass.

Let $\mathcal{S} \subseteq \text{Range}(\mathcal{M})$ consist of a single element which is the generator that returns $x_1$ with probability 1. Since 1 is the index of the element for which $\mathcal{D}_x$ and $\mathcal{D}_x$ differ, we have the following:

$$\mathbb{P}\left[\mathcal{M}(\mathcal{D}_x) \in \mathcal{S}\right] = 1$$
$$\mathbb{P}\left[\mathcal{M}(\mathcal{D}_y) \in \mathcal{S}\right] = 0$$

The definition of differential privacy is equivalent to the following inequality holding with probability $1 - \delta$ for every $\mathcal{S}' \subseteq \text{Range}(\mathcal{M})$ (Dwork & Roth, 2014):

$$\left|\log\left(\frac{\mathbb{P}\left[\mathcal{M}(\mathcal{D}_y) \in \mathcal{S}'\right]}{\mathbb{P}\left[\mathcal{M}(\mathcal{D}_x) \in \mathcal{S}'\right]}\right)\right| \leq \epsilon$$

Since when $\mathcal{S}' = \mathcal{S}$, this inequality only holds for $\epsilon = \infty$, $\mathcal{M}$ is not $(\epsilon, \delta)$-DP for finite $\epsilon, \delta$. $\square$

### B.3 Proof of Theorem 3.1

We first consider the marginal model system (5) in section 3 of the main text. We note that $V_m(\xi, \omega)$ and $U_m(\xi, \omega)$ are equivalent to $V(G, D)$ and $U(G, D)$ in (Zhao et al., 2017) for $x'$. Thus by Theorem 1 in (Zhao et al., 2017), it follows that if $(\xi^*, \omega^*)$ is a Nash equilibrium for (5), then $p_{\omega*} = p_{x'}$ almost everywhere and $V_m(\xi^*, \omega^*) = m$.

Now we consider the conditional model system (6) in section 3 of the main text. Using the same proof technique as for Theorem 1 in (Zhao et al., 2017) with a modification to account for our partitioning of $x$ into $\{x^*, x'\}$, we can show that $\mathcal{G}_\upsilon$ produces samples that are indistinguishable from the conditional distribution of $x^*$ given $x'$ in the real data.

First, we observe the following:

$$V_c(\gamma, \upsilon^*) = \int_x p_x(x)\mathcal{E}_\gamma(x^*|x')dx + \int_{z',x'} p_{z'}(z)p_{x'}(x')\left[m - \mathcal{E}_\gamma\left(\mathcal{G}_{\upsilon*}(z'|x')|x'\right)\right]^+ dz'dx' \tag{19}$$

$$= \int_x \left(p_{x^*|x'}(x^*|x')p_{x'}(x')\mathcal{E}_\gamma(x^*|x') + p_{\upsilon*}(x^*|x')p_{x'}(x')\left[m - \mathcal{E}_\gamma(x^*|x')\right]^+\right) dx \tag{20}$$

From Lemma 1 in the appendix of (Zhao et al., 2017), the analysis of the function $\varphi(y) = ay + b[m - y]^+$ shows:

(a) $\mathcal{E}_{\gamma^*}(x^*|x') \leq m$ almost everywhere. To verify, assume there exists a set of measure non-zero such that $\mathcal{E}_{\gamma^*}(x^*|x') > m$ and let $\mathcal{E}_{\tilde{\gamma}}(x^*|x') = \min(\mathcal{E}_{\gamma^*}(x^*|x'), m)$. Then $V_c(\tilde{\gamma}, \upsilon^*) < V_c(\gamma^*, \upsilon^*)$ so $(\gamma^*, \upsilon^*)$ is not a Nash equilibrium.

(b) $\varphi$ reaches its minimum in $m$ if $a < b$ and in 0 otherwise so $V_c(\gamma, \upsilon^*)$ reaches its minimum when we replace $\mathcal{E}_{\gamma^*}$ by these values. We thus observe the following:

$$V_c(\gamma^*, \upsilon^*) = m\int_x \mathbb{1}_{p_{x^*|x'}(x^*|x')p_{x'}(x')<p_{\upsilon*}(x^*|x')p_{x'}(x')} p_{x^*|x'}(x^*|x')p_{x'}(x')dx$$

$$+ m\int_x \mathbb{1}_{p_{x^*|x'}(x^*|x')p_{x'}(x')\geq p_{\upsilon*}(x^*|x')p_{x'}(x')} p_{\upsilon*}(x^*|x')p_{x'}(x')dx \tag{21}$$

$$= m\int_x \mathbb{1}_{p_{x^*|x'}(x^*|x')<p_{\upsilon*}(x^*|x')} p_{x^*|x'}(x^*|x')p_{x'}(x')dx$$

$$+ \left(1 - \mathbb{1}_{p_{x^*|x'}(x^*|x')<p_{\upsilon*}(x^*|x')}\right) p_{\upsilon*}(x^*|x')p_{x'}(x')dx \tag{22}$$

$$= m\int_x p_{\upsilon*}(x^*|x')p_{x'}(x')dx$$

$$+ m\int_x \mathbb{1}_{p_{x^*|x'}(x^*|x')<p_{\upsilon*}(x^*|x')} \left(p_{x^*|x'}(x^*|x') - p_{\upsilon*}(x^*|x')\right) p_{x'}(x')dx \tag{23}$$

$$= m + m\int_x \mathbb{1}_{p_{x^*|x'}(x^*|x')<p_{\upsilon*}(x^*|x')} \left(p_{x^*|x'}(x^*|x') - p_{\upsilon*}(x^*|x')\right) p_{x'}(x')dx \tag{24}$$

Since the second term in (24) above is non-positive $V_c(\gamma^*, \upsilon^*) \leq m$.

Now, putting the ideal conditional generator into the right side of $U_c(\gamma^*, \upsilon^*) \leq U_c(\gamma^*, \upsilon)$, we get:

$$\int_x p_{\upsilon*}(x^*|x')p_{x'}(x')\mathcal{E}_{\gamma^*}(x^*|x')\, dx \leq \int_x p_{x^*|x'}(x^*|x')\, p_{x'}(x')\, \mathcal{E}_{\gamma^*}(x^*|x')\, dx \tag{25}$$

Thus by (20) above, we get the following:

$$\int_x \left(p_{\upsilon*}(x^*|x')p_{x'}(x')\mathcal{E}_{\gamma^*}(x^*|x') + p_{\upsilon*}(x^*|x')p_{x'}(x')\left[m - \mathcal{E}_{\gamma^*}(x^*|x')\right]^+\right) dx \leq V_c(\gamma^*, \upsilon^*) \tag{26}$$

Since $\mathcal{E}_{\gamma^*}(x^*|x') \leq m$, we get $m \leq V_c(\gamma^*, \upsilon^*)$. Thus since $V_c(\gamma^*, \upsilon^*)$ is bounded above and below by $m$, we have $V_c(\gamma^*, \upsilon^*) = m$.

Now by (24) above we see that $V_c(\gamma^*, \upsilon^*) = m$ can only happen when $\int_x \mathbb{1}_{p_{x^*|x'}(x^*|x') < p_{\upsilon^*}(x^*|x')} dx = 0$ or equivalently $\int_x \mathbb{1}_{p_{x^*|x'}(x^*|x')p_{x'}(x') < p_{\upsilon^*}(x^*|x')p_{x'}(x')} dx = 0$. Since $p_{x^*|x'}(x^*|x')p_{x'}(x')$ and $p_{\upsilon^*}(x^*|x')p_{x'}(x')$ are probability densities for $x$, by Lemma 2 in the appendix of (Zhao et al., 2017) this is true if and only if $p_{x^*|x'}(x^*|x')p_{x'}(x') = p_{\upsilon^*}(x^*|x')p_{x'}(x')$ almost everywhere or equivalently $p_{x^*|x'}(x^*|x') = p_{\upsilon^*}(x^*|x')$ almost everywhere. Thus, we have $p_{\omega^*}p_{\upsilon^*} = p_{x'}p_{x^*|x'} = p_x$ almost everywhere.

## B.4 Proof of Theorem 3.2

We first consider the marginal model system (5) in section 3 of the main text. We note that $V_m(\xi, \omega)$ and $U_m(\xi, \omega)$ are equivalent to $V(G, D)$ and $U(G, D)$ in (Zhao et al., 2017) for $x'$. Thus by Theorem 2 in (Zhao et al., 2017), it follows that a Nash equilibrium for (5) exists and is characterized by $p_{\omega^*} = p_{x'}$ almost everywhere and there exists a constant $\nu_m \in [0, m]$ such that $\mathcal{E}_{\xi^*}(x') = \nu_m$ almost everywhere.

Now we consider the conditional model system (6) in section 3 of the main text. Using the same proof technique as for theorem 2 in the appendix of (Zhao et al., 2017), it follows that a Nash equilibrium for (6) exists and is characterized by $p_{\upsilon^*} = p_{x^*|x}$ almost everywhere and there exists a constant $\nu_c \in [0, m]$ such that $\mathcal{E}_{\gamma^*}(x^*|x') = \nu_c$ almost everywhere.

As in theorem 2 in (Zhao et al., 2017), the sufficient conditions are obvious. The necessary condition on $\mathcal{G}_{\upsilon^*}$ comes theorem 3.1. The necessary condition $\mathcal{E}_{\gamma^*}(x^*|x') \le m$ comes from the proof of theorem 3.1. Now we assume $\mathcal{E}_{\gamma^*}(x^*|x')$ is not constant and arrive at a contradiction.

If $\mathcal{E}_{\gamma^*}(x^*|x')$ is not constant, then there is a constant $C$ and set $\mathcal{S}$ of non-zero measure such that $\forall x \in \mathcal{S}$, $\mathcal{E}_{\gamma^*}(x^*|x') \le C$ and $\forall x \notin \mathcal{S}$, $\mathcal{E}_{\gamma^*}(x^*|x') > C$. We can choose $\mathcal{S}$ such that there exists $\mathcal{S}' \subset \mathcal{S}$ of non-zero measure such that $p_{x^*|x'}(x^*|x') > 0$ on $\mathcal{S}$. We can build a generator $\mathcal{G}_{\tilde{\upsilon}}$ such that $p_{\tilde{\upsilon}}(x^*|x') \le p_{x^*|x'}(x^*|x')$ on $\mathcal{S}$ and $p_{\tilde{\upsilon}}(x^*|x') < p_{x^*|x'}(x^*|x')$ on $\mathcal{S}'$. We get the following:

$$
\begin{aligned}
U_c(\gamma^*, \upsilon^*) - U_c(\gamma^*, \tilde{\upsilon}) &= \int_x \left( p_{x^*|x'}(x^*|x') - p_{\tilde{\upsilon}}(x^*|x') \right) p_{x'}(x') \mathcal{E}_{\gamma^*}(x^*|x') dx \\
&= \int_x \left( p_{x^*|x'}(x^*|x') - p_{\tilde{\upsilon}}(x^*|x') \right) p_{x'}(x') \left( \mathcal{E}_{\gamma^*}(x^*|x') - C \right) dx \\
&= \int_{\mathcal{S}} \left( p_{x^*|x'}(x^*|x') - p_{\tilde{\upsilon}}(x^*|x') \right) p_{x'}(x') \left( \mathcal{E}_{\gamma^*}(x^*|x') - C \right) dx \\
&\quad + \int_{\mathbb{R}^d \backslash \mathcal{S}} \left( p_{x^*|x'}(x^*|x') - p_{\tilde{\upsilon}}(x^*|x') \right) p_{x'}(x') \left( \mathcal{E}_{\gamma^*}(x^*|x') - C \right) dx \\
&> 0
\end{aligned}
$$

Then $U_c(\gamma^*, \upsilon^*) > U_c(\gamma^*, \tilde{\upsilon})$ so $(\gamma^*, \upsilon^*)$ is not a Nash equilibrium.

## B.5 Proof of Lemma 3.1

*Proof.*

We can write $\hat{\mathbb{S}}_{l,n}$ as follows:

$$
\begin{aligned}
\hat{\mathbb{S}}_{l,n}(p, q) &= \frac{(\lfloor n/2 \rfloor - 1) \left[ \sum_{i=1}^{\lfloor n/2 \rfloor} u_q(x_{2i-1}, x_{2i}) \right]^2}{\lfloor n/2 \rfloor \sum_{i=1}^{\lfloor n/2 \rfloor} u_q(x_{2i-1}, x_{2i})^2} \\
&= \frac{(\lfloor n/2 \rfloor - 1)}{\sum_{i=1}^{\lfloor n/2 \rfloor} u_q(x_{2i-1}, x_{2i})^2} \left( \frac{\left[ \sum_{i=1}^{\lfloor n/2 \rfloor} u_q(x_{2i-1}, x_{2i}) \right]}{\sqrt{\lfloor n/2 \rfloor}} \right)^2 \\
&= g(x)
\end{aligned}
$$

where

$$x = \frac{\left[ \sum_{i=1}^{\lfloor n/2 \rfloor} u_q(x_{2i-1}, x_{2i}) \right]}{\sqrt{\lfloor n/2 \rfloor}} \qquad g(x) = \frac{\lfloor n/2 \rfloor - 1}{\sum_{i=1}^{\lfloor n/2 \rfloor} u_q(x_{2i-1}, x_{2i})^2} x^2$$

From (Liu et al., 2016), the above quantity $x = \lfloor n/2 \rfloor^{-\frac{1}{2}} \sum_{i=1}^{\lfloor n/2 \rfloor} u_q(x_{2i-1}, x_{2i})$ has an asymptotic $\mathcal{N}\left(0, \sigma_{u_q}^2\right)$ distribution under the null hypothesis $p = q$. Thus, since $\frac{\lfloor n/2 \rfloor - 1}{\sum_{i=1}^{\lfloor n/2 \rfloor} u_q(x_{2i-1}, x_{2i})^2} > 0$ and $x^2$ is symmetric and monotonically decreasing for $x \leq 0$ and monotonically increasing for $x \geq 0$, using the change of variables formula, $\hat{\mathbb{S}}_{l,n}$ has the following density function:

$$f_{\hat{\mathbb{S}}_{l,n}}(y) = 2 f_x \left( g^{-1}(y) \right) \frac{dg^{-1}(y)}{dy}$$

where

$$g^{-1}(y) = \sqrt{\frac{\sum_{i=1}^{\lfloor n/2 \rfloor} u_q(x_{2i-1}, x_{2i})^2}{\lfloor n/2 \rfloor - 1} y} \qquad \frac{dg^{-1}(y)}{dy} = \frac{1}{2} \sqrt{\frac{\sum_{i=1}^{\lfloor n/2 \rfloor} u_q(x_{2i-1}, x_{2i})^2}{(\lfloor n/2 \rfloor - 1) y}}$$

Thus, we can write the density function for $\hat{\mathbb{S}}_{l,n}$ as follows:

$$\begin{aligned}
f_{\hat{\mathbb{S}}_{l,n}}(y) &= 2 f_x \left( g^{-1}(y) \right) \frac{dg^{-1}(y)}{dy} \\
&= 2 \frac{1}{\sqrt{2\pi\sigma_{u_q}^2}} \exp\left( \frac{-\left(g^{-1}(y)\right)^2}{2\sigma_{u_q}^2} \right) \frac{dg^{-1}(y)}{dy} \\
&= 2 \frac{1}{\sqrt{2\pi\sigma_{u_q}^2}} \exp\left( \frac{-\frac{\sum_{i=1}^{\lfloor n/2 \rfloor} u_q(x_{2i-1}, x_{2i})^2}{\lfloor n/2 \rfloor - 1} y}{2\sigma_{u_q}^2} \right) \frac{1}{2} \sqrt{\frac{\sum_{i=1}^{\lfloor n/2 \rfloor} u_q(x_{2i-1}, x_{2i})^2}{(\lfloor n/2 \rfloor - 1) y}}
\end{aligned}$$

Now plugging in the sample variance estimator for $\sigma_{u_q}^2$, we get the following

$$\begin{aligned}
f_{\hat{\mathbb{S}}_{l,n}}(y) &= 2 \frac{1}{\sqrt{2\pi\sigma_{u_q}^2}} \exp\left( \frac{-\frac{\sum_{i=1}^{\lfloor n/2 \rfloor} u_q(x_{2i-1}, x_{2i})^2}{\lfloor n/2 \rfloor - 1} y}{2\sigma_{u_q}^2} \right) \frac{1}{2} \sqrt{\frac{\sum_{i=1}^{\lfloor n/2 \rfloor} u_q(x_{2i-1}, x_{2i})^2}{(\lfloor n/2 \rfloor - 1) y}} \\
&= \frac{1}{\sqrt{2\pi y}} \exp\left( \frac{-y}{2} \right) \\
&= \frac{1}{\Gamma(1/2) \, 2^{1/2}} y^{\frac{1}{2} - 1} \exp\left( \frac{-y}{2} \right)
\end{aligned}$$

which is the $\chi_1^2$ density function $\qquad \square$

## B.6 Proof of Theorem 3.3

*Proof.*

By Lemma 3.1, $\hat{\mathbb{S}}_{l,n}$ has an asymptotic $\chi_1^2$ distribution under the null hypothesis $p = q$. Thus, the quantity $F_{\chi_1^2}^{-1}(1 - \alpha) > \hat{\mathbb{S}}_{l,n}(p_{x'}, p_\xi)$ if and only if $\hat{\mathbb{S}}_{l,n}(p_{x'}, p_\xi)$ falls outside of the rejection region of a level-$\alpha$ test for the null hypothesis $\{x'\}_{i=1}^n \sim p_\xi$ for a minibatch $\{x'\}_{i=1}^n$.

Now consider the possible parameter settings for $\xi$ and partition them into $\eta_{reject}$ and $\eta_{fail}$, where $\eta_{reject}$ consists of all settings of $\xi$ such that $H_0 : \{x'\}_{i=1}^n \sim p_\xi$ is rejected using the above test and $\eta_{fail}$ consists of all

other settings. Define the following quantities (where $L_\xi(x_i', z_i)$ and $P_\xi^L(x_i', z_i)$ are defined as in Algorithm 1):

$$\xi_{reject}^* = \underset{\xi \in \eta_{reject}}{\arg\min} \left( \sum_{i=1}^n \left[ L_\xi(x_i', z_i) + P_\xi^L(x_i', z_i) \right] + \lambda_\alpha \left[ F_{\chi_1^2}^{-1}(1-\alpha) - \hat{\mathbb{S}}_{l,n}(p_{x'}, p_\xi) \right]^+ \right)$$

$$\xi_{fail}^* = \underset{\xi \in \eta_{fail}}{\arg\min} \left( \sum_{i=1}^n \left[ L_\xi(x_i', z_i) + P_\xi^L(x_i', z_i) \right] + \lambda_\alpha \left[ F_{\chi_1^2}^{-1}(1-\alpha) - \hat{\mathbb{S}}_{l,n}(p_{x'}, p_\xi) \right]^+ \right)$$

From the above observation from Lemma 3.1, we have the following:

$$\lambda_\alpha \left[ F_{\chi_1^2}^{-1}(1-\alpha) - \hat{\mathbb{S}}_{l,n}(p_{x'}, p_{\xi_{reject}^*}) \right]^+ = 0$$

$$\lambda_\alpha \left[ F_{\chi_1^2}^{-1}(1-\alpha) - \hat{\mathbb{S}}_{l,n}(p_{x'}, p_{\xi_{fail}^*}) \right]^+ > 0$$

Define $L_\xi$ as follows:

$$L_\xi = \sum_{i=1}^n \left[ L_\xi(x_i', z_i) + P_\xi^L(x_i', z_i) \right]$$

Now assume that for some iteration of the outer loop of algorithm 1, $\xi$ converges to a global optimum $\xi^*$ in the inner loop. If $\xi^* = \xi_{fail}^*$, then we have the following:

$$L_{\xi_{fail}^*} + \lambda_\alpha \left[ F_{\chi_1^2}^{-1}(1-\alpha) - \hat{\mathbb{S}}_{l,n}(p_{x'}, p_{fail}^*) \right]^+ < L_{\xi_{reject}^*} + \lambda_\alpha \left[ F_{\chi_1^2}^{-1}(1-\alpha) - \hat{\mathbb{S}}_{l,n}(p_{x'}, p_{\xi_{reject}^*}) \right]^+$$

$$L_{\xi_{fail}^*} + \lambda_\alpha \left[ F_{\chi_1^2}^{-1}(1-\alpha) - \hat{\mathbb{S}}_{l,n}(p_{x'}, p_{fail}^*) \right]^+ < L_{\xi_{reject}^*}$$

$$\lambda_\alpha \left[ F_{\chi_1^2}^{-1}(1-\alpha) - \hat{\mathbb{S}}_{l,n}(p_{x'}, p_{fail}^*) \right]^+ < L_{\xi_{reject}^*} - L_{\xi_{fail}^*}$$

Thus, there exists $\lambda_{\alpha^*} > \lambda_\alpha$, such that if we replace $\lambda_\alpha$ with $\lambda_{\alpha^*}$, the above inequality would be reversed and we would have $\xi^* = \xi_{reject}^*$. Thus, for sufficiently large $\lambda_\alpha$, when Algorithm 1 is trained to convergence in the inner loop and reaches a global optimum, we can reject $H_0 : \{x'\}_{i=1}^n \sim p_\xi$ using the above test.

Now, let $p_\omega$ be a representation for the generative distribution of $\mathcal{G}_\omega$ When the inner loop of Algorithm 1 is trained to convergence and reaches a global optimum with sufficiently large $\lambda_\alpha$, $\mathcal{G}_\omega$ observes $x'$ indirectly through $\mathcal{E}_{\xi_{reject}^*}$. From the data processing inequality, we have

$$D_{KL}(p_{x', \xi_{reject}^*} || p_{x'} p_{\xi_{reject}^*}) \geq D_{KL}(p_{x', \omega} || p_{x'} p_\omega)$$

Thus, rejection of the level-$\alpha$ test for $\{x'\}_{i=1}^n \sim p_\xi$ entails rejection of the level-$\alpha$ test for $\{x'\}_{i=1}^n \sim p_\omega$. $\qquad \square$

## C  Dataset Descriptions

### C.1  1-D Gaussian Mixture Models to Compare the Distributions of $\hat{\mathbb{S}}_{l,n}$ and $\hat{\mathbb{S}}_l$

Below are the randomly chosen parameters for the 1-D Gaussian Mixture Model used to evaluate the distributions of $\hat{\mathbb{S}}_l$ and $\hat{\mathbb{S}}_l$ when $p = q$ and $p \neq q$.

| Mixture Weight | $\mu$ | $\sigma^2$ |
|---|---|---|
| 0.2 | 6.9734 | 0.2622 |
| 0.2 | 7.1579 | 0.9046 |
| 0.2 | 2.4358 | 0.4533 |
| 0.2 | 9.2579 | 0.0773 |
| 0.2 | 3.0420 | 0.7823 |

## C.2 Kaggle Credit Card Fraud

This dataset consists of 284,807 credit card transactions made by European cardholders which occurred during two days in September 2013 (Pozzolo et al., 2015). Each transaction record includes 28 continuous variables which result from a PCA transformation (the original variables as well as descriptions are unavailable due to confidentially) as well as the time and amount of the transaction and a binary indicator of whether the transaction was fraudulent. The dataset is highly imbalanced; fraudulent transactions make up 0.172% of all transactions. We exclude the time variable in our experiments, treating each transaction as occurring independent of all previous transactions and standardize the variables before training. The license for this dataset is available at `https://opendatacommons.org/licenses/dbcl/1-0/`.

## C.3 MIMIC-III Critical Care

MIMIC-III (Medical Information Mart for Intensive Care) is a publicly available (subject to approval) electronic health care record (EHR) dataset that is widely used as a benchmark (Johnson et al., 2016). It includes data associated with 53,423 distinct hospital admissions to critical care units between 2001 and 2012, e.g. vital signs, medications, laboratory measurements, observations and notes charted by care providers, fluid balance, procedure codes, diagnostic codes, imaging reports, hospital length of stay and survival data. The license for this dataset is available at `https://physionet.org/content/mimiciii/view-license/1.4/`. Please see `https://mimic.mit.edu/iii/about/` and (Johnson et al., 2016) for details regarding personally identifiable information and obtaining consent as well as requirements for researchers to use MIMIC-III.

For dataset construction, we follow the same procedure as in (Choi et al., 2017) and (Xie et al., 2018), focusing on only the ICD-9 codes for each patient. ICD-9 codes (International Statistical Classification of Diseases and Related Health Problems) identity a particular disease or health problem that a patient has been diagnosed with during a hospital stay. We group ICD-9 codes according to their first 3 digits, which results in 1071 possible diagnoses for each hospital stay. We then construct longitudinal health records for each patient $\{0,1\}^{1071}$ such that each ICD-9 code group is coded as 1 or 0 if the patient received or never received the diagnosis during any recorded hospital stay, respectively. This results in 46,520 individual longitudinal patient health records which each contain 1071 binary attributes.

# D   Experimental Settings and Additional Results

## D.1   Hyperparameters

Below are the hyperparameters used in all experiments with PPEM models:

$m = 10$
$\lambda_\alpha = 10$
$\lambda_D = 1$
Training epochs $= 100$
Minibatch size $= 128$
Number of energy model iterations per generator iteration $= 5$
MLP layer dimensions (all networks) $= 256$
Latent dimensions (both generators) $= 128$
Generators learning rate $= 5e^{-4}$
Energy models learning rate $= 1e^{-3}$
$\alpha$-level $= 0.05$
$(\epsilon, \delta) = (1, n^{-1})$

For DP-GAN and PATE-GAN, we use the public implementation `https://github.com/BorealisAI/private-data-generation` with the default settings.

## D.2   Environment

All experiments were run using a single NVIDIA T4 GPU.

### D.3 KSD and MMD Test Statistic Values

Below are the KSD test statistic values and $p$-values resulting from the bootstrap test to evaluate fitness of the real attributes which require $\alpha$-LMDP protection to the learnt marginal energy distribution.

| Kaggle Credit Card Fraud | | |
|---|---|---|
| | KSD | $p$-value |
| PPEM (2) | 220.2 | 0.00 |
| PPEM (5) | 60.86 | 0.00 |
| PPEM (2) + DP | 131.9 | 0.00 |
| PPEM (5) + DP | 213.6 | 0.00 |

| MIMIC-III EHRs | | |
|---|---|---|
| | KSD | $p$-value |
| PPEM (50) | 77.16 | 0.00 |
| PPEM (200) | 162.9 | 0.00 |
| PPEM (50) + DP | 18.48 | 0.00 |
| PPEM (200) + DP | 241.7 | 0.00 |

Below are the MMD test statistic values and $p$-values to evaluate whether the samples generated from each model for the attributes requiring $\alpha$-LMDP protection are from the same distribution as the real private attributes.

| Kaggle Credit Card Fraud | | |
|---|---|---|
| | MMD | $p$-value |
| WGAN (No Privacy) | 0.0417 | 0.15 |
| EnergyGAN (No Privacy) | 0.5891 | 0.00 |
| SteinGAN (No Privacy) | 0.1445 | 0.00 |
| PPEM (2) | 0.0530 | 0.02 |
| PPEM (5) | 0.0667 | 0.00 |
| PPEM (2) + DP | 0.0571 | 0.01 |
| PPEM (5) + DP | 0.1570 | 0.00 |
| DP-GAN | 0.0968 | 0.00 |
| PATE-GAN | 0.1075 | 0.00 |

### D.4 Comparison of PPEMs with $\alpha$-LMDP to DP applied only to $x'$

We include an additional experiment where we compare PPEMs with $\alpha$-LMDP to only applying Differential Privacy to the attributes $x'$ to directly compare the impact to fidelity for these two different privacy penalties. To train with DP applied to only $x'$, we using the same model and training procedure as for PPEMs, but we do not include the KSD-based penalty and perturb gradients in the marginal model, but not in the conditional model. We apply this procedure to the Kaggle Credit Card Fraud dataset and compare the results to PPEMs with $\alpha$-LMDP. Below, we show the KSD test results on the private attributes and the classification and regression results as in the experiments in the main paper when $x'$ includes 2 and 5 attributes.

|  | KSD | $p$-value | AU-ROC | MSE | $R^2$ |
|---|---|---|---|---|---|
| (2) 0.05-LMDP | 220.2 | 0.00 | 0.922 | 0.184 | 0.822 |
| (2) $(1, n^{-1})-$DP | 7.181 | 0.00 | 0.903 | 0.602 | 0.421 |
| (5) 0.05-LMDP | 60.86 | 0.00 | 0.948 | 0.196 | 0.812 |
| (5) $(1, n^{-1})-$DP | 1.212 | 0.00 | 0.891 | 0.720 | 0.307 |

We observe that training with both penalties results in models which satisfy $\alpha$-LMDP, but the regression and classification performance is stronger for the PPEMs trained to ensure $\alpha$-LMDP, indicating the KSD-based penalty has a weaker impact on the fidelity of the resulting models than DP. These results argue that PPEMs trained to ensure $\alpha$-LMDP have a favorable fidelity vs. privacy tradeoff.

