# OpenReview forum: "Privacy-Preserving Energy-Based Generative Models for Marginal Distribution Protection"
_TMLR — Accepted by TMLR_

### Review · Reviewer_mZ5y · 2023-01-05

**Summary Of Contributions:**


This paper proposes a new class of differential privacy which protects against the discovery of certain marginal distributions in the data generated by a generative model. These marginal distributions can represent protected demographic or business attributes that the model designer does not want leaked through examining the model output.

The paper proposes a method for training a certain class (GAN-based) of generative models in a manner that preserves this new differential privacy definition, and demonstrates its effectiveness on a number of tabular benchmarks.


**Audience:**

Yes

**Broader Impact Concerns:**

It would have been interesting to reflect on the fact that this setting basically enforces that the data generated doesn't match the distribution of a protected category. While it may be desirable in the context of preserving privacy, the notion that data that will be used by a downstream application to be inherently non-reflective of the real distribution of a protected attribute is potentially concerning. For example: if the attribute is ethnicity, may it induce the model to generate data that is more or less diverse than reality? How would the end-user be able to tell that they're not exploiting that biased protected category distribution? This is generally not an issue differential privacy has to reason about, because it concerns itself with individuals, but in this setting, it may matter quite a bit.

**Claims And Evidence:**

Yes

**Requested Changes:**

Nits:
- The title of the paper does it a disservice by not mentioning at all the 'Marginal Distribution Privacy' concept, which is much more central to the contribution than the solution happening to be 'energy-based'. I wouldn't request to change it per se, but encourage the authors to reconsider it.
- In the first equation in Section 2, the symbol P() is not defined. People may reasonably be expected to understand it's a probability distribution, but unless they are very versed in differential privacy, they may not immediately know a distribution over what. Should be clarified.
- '\alpha-Level' is not defined until Definition 2.2: 'a level-\alpha goodness of fit test' - even that definition is not entirely illuminating until much later in the paper where is is made more clear that \alpha is meant as what's usually called the '\epsilon' confidence level of the test. Qualifying what this \alpha is should be unambiguous and happen much sooner.


**Strengths And Weaknesses:**

Disclaimers:
- while familiar with the GAN and differential privacy literature, this reviewer is not an expert at either.
- in particular, examining the proofs in appendix B, section 5+ is beyond my level of mathematical understanding of the problem.

Strengths:
- Relevant problem, which adds a new conceptual tool to the portfolio of differential privacy with strong potential for practical utility.
- Well written paper, clear and detailed.
- Elegant, general approach to solving for this new DP definition.

Weaknesses:
- I would have appreciated some commentary on how this may apply to generative models that are not GAN. In many applications, Diffusion Models or even Transformer models trained on a generative objective are completely replacing GANs due to their improves simplicity and ease of training.
- There is no limitation section. I would have appreciated to read about things that were attempted and didn't work, or any identified limitation of this work beyond the superficial treatment given to it in the Conclusion.

---

> ### Author Response · Authors · 2023-04-15
> **Response to Reviewer mZ5y**
>
> We thank the reviewer for their time and helpful comments and suggestions. We agree with the reviewer’s suggested revisions. Below we provide responses and discussion along with our specific planned revisions - if any do not fully address a concern raised, we kindly ask if the reviewer could indicate further revisions requested.
>
> **Responses:**
> 1. *Generative models that are not GANs*  -  Our choice of an EnergyGAN-style architecture, partitioned so the protected marginal distributions can be trained independently of the conditional distribution of the other attributes is motivated by our use of the KSD for testing goodness of fit, which requires access to the score function (which we get for free when using an energy model). It might be possible to use other generative models like diffusion models, which are becoming more popular, but it would require a reformulation of the models so that scores could be computed (and so that this could be done efficiently). It is not clear exactly how such a reformation would look, but we will add a limitation section with commentary to explain this.
> 2. *Limitations section* - we agree the paper would benefit from this. We will add this to the paper and address several specific limitations (see [A] below).
> 3. *Title*  -  We agree the title does not mention this key aspect of our paper (see [B] below)
> 4. *Impacts to potential downstream applications / bias to protected attributes* - We thank the reviewer for raising this important broader impact point, related the point raised by Reviewer KKXi on the relation of this approach to fair generative modeling. We will add a broader impact section and address this alongside Reviewer KKXi’s point.
>
> **Planned Revisions**
>
> A. We will add a limitations section to the paper. Specifically, we will include (i) the limitation mentioned in response 2 in our above official comment to Reviewer nErC, (ii) the fact that our approach requires specifying only a single subset of marginals to protect (we would like to extend it to include multiple sets), (iii) the design of our methodology and it’s relation to the KSD makes it more difficult to apply to generative architectures that are not energy models (the reviewer’s point discussed in [1] above).
>
> B. To reflect that marginal distribution privacy aspect of the paper we will retitle our paper “Privacy-Preserving Energy-Based Generative Models for Marginal Distribution Protection”
>
> C. We will clearly define $P()$ and $\alpha$ before their usage as requested by the reviewer.
>
> D. We will add a broader impact section and include the reviewer's point mentioned above in [4] alongside Reviewer KKXi’s point relating our approach to fair generative modeling. Specifically, we will point out that if protected attributes for fairness, such as ethnicity, are included in the marginal distribution, the resulting generative distribution for these attributes may be less diverse and this may result in downstream models that are less fair (or more diverse and more fair as there is no guarantee either way). We will emphasize that data owners must take this into account, consider any bias and fairness concerns alongside privacy concerns and be cognizant of their interaction if using our approach.

---

> > ### Comment · Reviewer_mZ5y · 2023-05-03
> > **Thank you for the replies.**
> >
> > No further concern. This is a strong contributions.

---

### Review · Reviewer_KKXi · 2023-01-11

**Summary Of Contributions:**

The paper considers a new formulation of privacy in generative models, in which we are interested in protecting the marginal distribution of some protected attributes of the generated data. They term this **$\alpha$-Level Marginal Distribution Privacy** ($\alpha$-LMDP), and they argue it can be important for, e.g., financial institutions interested in sharing fake generated data while also hiding aggregate statistics on their consumers.

The way they approach the problem is to decompose the original distribution $p(x)$ into a distribution on protected attributes $p(x')$ and a conditional distribution $p(x^* | x')$ on the remaining attributes. These are implemented as two separate generators, which are trained with an energy-based criterion that interprets the discriminator of the GAN as an energy function (EnergyGAN). They show that under this interpretation, adding the $\alpha$-LMDP term as regularization can be done efficiently by using the score function of the energy function and the Kernel Stein Discrepancy criterion.

**Audience:**

Yes

**Broader Impact Concerns:**

As the paper is discussing ways to mitigate the impact of releasing generative models in certain contexts, I believe there are no specific concerns to discuss.

**Claims And Evidence:**

Yes

**Requested Changes:**

* (Optional) Reformulate the discussion as per point 1) above.
* I am curious about the section "3.4 Optimality of the solution". Upon first reading, it appears to be very similar to the optimality proof from the EnergyGAN paper, by considering a factorization of $p(x)$ in two parts. While I am not suggesting to remove this proof, it would be interesting to discuss if there are technical difficulties in extending the EnergyGAN proof here, or if this is, in some sense, a simple extension.
* Clarify the experimental evaluation as per points (2), (3), and (4) above.
* Amend the claims of the paper in accordance to point (5) above.
* I would also suggest to discuss the relation of this paper with fair generative modeling, where the aim is to generate data whose distribution w.r.t. (possibly unobserved) latents is as uniform as possible.

**Strengths And Weaknesses:**

**Strengths:**
- The paper is well written and easy to follow.
- The motivation is clear and the proposed $\alpha$-LMDP criterion is novel to my knowledge (although I am not an expert on the privacy-preserving literature).
- The theoretical additions to the paper are also good (showing that DP and $\alpha$-LMDP are, in a sense, orthogonal, and extending the EnergyGAN optimality proof).

**Weaknesses:**
1) I think the relation between GANs and energy models is not very clear from the paper. In particular, (6) and (7) are the training criterion of EnergyGAN, which interpretes the discriminator of the GAN as an energy function. This is also stated in Section 4.3 in the paper. I think it would be easier for reading to move this to *3.1 Background* and simplify the description in *3.2 Proposed Model*.
2) I am not sure I fully understand the experimental evaluation. In particular, in Table 1 almost every method has a p-value for MMD < 0.05. Does this mean that the original problem was not very significant to begin with? There is literature [1] on generating data with realistic marginal distribution (which is the opposite problem as to here) which also points to this as potential conclusion.
3) I am also curious about why the proposed PPEM can improve the AUROC compared to the EnergyGAN while being almost identical in training.
4) Finally, why is the MMD test not replicated also for the MIMIC dataset?
5) The sentence "PPEMs are the first energy-based models with privacy-preserving properties" is quite strong. Again, I am not an expert but a very quick search found [1] and [2] on Scholar (diffusion models being strongly linked with energy-based models).

[1] https://arxiv.org/abs/2105.06907
[2] https://openaccess.thecvf.com/content/CVPR2022/papers/Chen_DPGEN_Differentially_Private_Generative_Energy-Guided_Network_for_Natural_Image_Synthesis_CVPR_2022_paper.pdf
[3] https://arxiv.org/abs/2210.09929

---

> ### Author Response · Authors · 2023-04-15
> **Response to Reviewer KKXi**
>
> We thank the reviewer for their time and helpful comments and suggestions. We agree with the suggested revisions. Below we provide specific responses to questions along with our planned revisions - if any do not fully address a concern raised, we kindly ask if the reviewer could indicate further revisions requested.
>
> **Responses:**
>
> 1. *Why does the proposed PPEM improve AUROC compared to EnergyGAN?*  -  We provide some commentary on this in the second paragraph of 5.2.2 (and hint at it earlier in 3.7), but will modify the text to address this explicitly. EnergyGAN, when applied to tabular datasets, suffers from unstable training and thus generally has poor performance on both of the datasets tested (EnergyGAN, when proposed, was applied only to image datasets, which contain more structure). Hence, WGAN is a better baseline to compare to. Initially, we left EnergyGAN out of the results to avoid this confusion since EnergyGAN's results are not very meaningful for these datasets, but a reviewer of a previous version of the paper requested its inclusion for completeness.
>
> 2. *Almost every method has a p-value less than .05*  -  The problem is not trivial. While the reviewer correctly notes that that the MMD is less than .05 in *almost* all cases, crucially it is *greater* than .05 for WGAN which is the only meaningful non-private baseline shown (see [1] above regarding the results EnergyGAN). This demonstrates that simply using a GAN does not protect marginal distributions. While the differentially private GANs (DP-GAN and PATE-GAN) do result in MMDs less than 0.05, this is not theoretically guaranteed, i.e. it is possible to protect the privacy of individuals in the dataset, but not the marginal distributions. Furthermore, the MMD is less than 0.05 for DP-GAN and PATE-GAN simply due to the fact that (as observed in the literature) DP (even at relatively low levels) has a significant negative impact on fidelity, i.e. marginal distribution protection only results from the poor fidelity - this is evidenced by the low AUROC scores for these models. In contrast, the PPEM results indicate marginal distribution protection while maintaining high fidelity.
>
> 3. *Why is the MMD not included for the MIMIC dataset?*  -  The MMD is generally applied to continuous data, where there are universal kernels defined which guarantee the theoretical properties of the MMD, e.g. the Gaussian and Laplace kernels. The MIMIC dataset we use, however, consists of all binary variables.
>
> 4. *"PPEMs are the first energy-based models with privacy-preserving properties"*  -  We thank the reviewer for providing this reference (which came out after we wrote the original version of the paper). We will remove this statement from the text.
>
> 5. *Optimality proof and relation to EnergyGAN*  -  Generally speaking, the proof is a relatively simple extension where we simply must account for the partioning of $x$ into $x'$ and $x^*$. We indicate this in the proof itself, but we will add this to the text.
>
> 6. *Relation to fair generative modeling*  -  We thank the reviewer for bringing this point up, which is important and related to the concern brought up by Reviewer mZ5y. We will add this to a broader impact section and address both reviewer's points together in this section (see [E] below)
>
> **Planned Revisions**
>
> A. To make the relationship between GANs, energy models, and EnergyGAN more clear, we will introduce EnergyGAN in section 3.1 as suggested by the reviewer.
>
> B. We will add the commentary in 5 above regarding the optimality proof and its relation to the proof from EnergyGAN to the main text.
>
> C. We will add the above responses 1, 2 and 3 to the text to address the points 2, 3 and 4 referenced above in the reviewer's requested changes.
>
> D. We will remove the statement that PPEMs are the first energy-based models with privacy-preserving properties from the paper.
>
> E. We will add a broader impact section and discuss the relation of this approach to fair generative modeling alongside the point brought up by Reviewer mZ5y. Specifically, we will clarify that while our approach, as a result of protecting marginal distributions, results in generative distributions different from the original distribution, it does not guarantee uniformity of these distributions and hence should not be applied to a marginal distribution consisting of protected attributes with the hope that the resulting generated data would be fair as our approach does not guarantee this outcome.

---

> > ### Comment · Reviewer_KKXi · 2023-04-18
> > **Answer to the authors**
> >
> > Many thanks for the comprehensive answers, I have no additional concerns on the manuscript.

---

### Review · Reviewer_nErC · 2023-04-03

**Summary Of Contributions:**

The paper considers privacy of marginal generative distribution behind some attributes of the data. This differs from differential privacy (DP) that considers privacy of an individual and is close to the recent work on confidentiality and privacy of dataset properties and distributions.
The authors propose a definition to capture the new notion of privacy based on goodness of fit test and relies on hypothesis testing. The authors show that their definition is orthogonal to DP (as one would expect).
The paper proposes a method for generating synthetic data that reproduces data close to the data based on non-protected attributes (using energy and GAN-like method).
The authors also provide experiments based on two datasets and evaluate privacy of their method using the goodness of fit test.

**Audience:**

Yes

**Claims And Evidence:**

No

**Requested Changes:**

The paper is overall sound wrt the claims it makes and it considers an interesting problem. If it was not in the area of privacy I believe its definition and techniques would be fine as in average case the method would work. Unfortunately privacy is a complex topic and making assumptions about what an adversary can do has been shown error prone (e.g., k-anonymity). In particular, the paper defends against an adversary that when trying to detect if data is distributed according to some distribution or not, will only do a goodness of fit test. This is a strong assumption as adversary could test for other properties and infer something about attribute distribution, thereby, leaking the information that needs to be protected. Similarly the method is based on training a neural network which is not optimal, hence, even the method itself may not satisfy the original definition (even if the definition was a strong privacy definition). The authors could consider their definition in view of other definitions to capturing dataset privacy or how one can considers DP protection in the hypothesis testing context:
"Hypothesis Testing Interpretations and Renyi Differential Privacy" by Balle et al AISTATS 2020
"Attribute Privacy: Framework and Mechanisms" by Zhang et al. FACCT 2022
"Formalizing and Estimating Distribution Inference Risks" by Suri and Evans PETS 2022

**Strengths And Weaknesses:**

Strengths:
- interesting problem, different from differential privacy
- the approach is interesting
- overall the paper is well-written
- experiments are comprehensive

Weaker ponts:
- privacy definition is not close to the strength of commonly accepted privacy definitions
- though the method is interesting, it protects against a very specific adversary (doing hypothesis testing)
- the definition and method are dataset dependant (not mechanism) which is not considered strong for data protection in DP

---

> ### Author Response · Authors · 2023-04-15
> **Response to Reviewer nErC**
>
>
>
> We thank the reviewer for their time and helpful comments and suggestions. We generally agree with both the positive and negative points raised and the requested changes, which we will make. Below we provide specific responses to points raised and planned revisions. If these do not fully address a concern raised, we kindly ask if the reviewer could indicate further revisions requested.
>
> **Responses:**
>
> 1. *Weaker notion of privacy than DP for this context and not mechanism-based*  -  We agree that our privacy definition would be considered weak in relation to DP and are also aware that this may result in negative perceptions from the privacy community. However, we believe the definition and related methodology provide a solution to an important practical problem in industry that is not well address by DP and related notions and our approach has a favorable set of tradeoffs compared to related work (see [4] below). Our goal is to clearly state the differences and limitations and address these questions head on. We focus on this at the end of section 2, but if there are any specific suggestions to better address these reactions (or at a different point in the paper), we're happy to include these in our revisions.
>
> 2. *Assuming what an adversary can do can be problematic, e.g. k-anonymity*  -  This is a sound concern. While our setup is different in multiple ways from the one in which k-anonymity was applied, which we believe partially mitigates this concern, we agree this limitation could be better acknowledged in the paper, which we plan to do in an added broader impact section (see [B] below). Specifically, our context is different than k-anonymity because the entity whose privacy would be violated is the data owner rather than the individuals who make up the dataset (whose privacy can still be protected using DP or a related notion) and the data owner is the best entity to determine what aspects of the dataset at the aggregate level must be protected from an adversary due to competitive concerns, provided the data owner is fully aware of the tradeoffs of different privacy notions.
>
> 3. *Training a neural network*  -  We chose a neural network because it does not make strong assumptions about the data distribution. We believe this is an advantage of our approach compared to related methods (see [A] below) both in terms of the fidelity of the generated data as well as making assumptions about protecting the privacy of the data distribution.
>
> 4. *Relationship to papers cited*  -  We agree the papers listed (which mostly came out after our paper was originally written) are important related papers which should be added. We will revise the paper to include them as related work and discuss the relation to our proposed privacy notion (see [A] below)
>
> **Planned Revisions:**
>
> A. We will add the papers listed by the reviewer to the related work and discuss them in relation to our proposed privacy notion, particularly focusing on Zhang et al., 2022, which is the closest to our proposed method. Specifically, we will point out the tradeoffs that while Zhang et al. 2022 (i) offers a privacy notion that is stronger and closer to DP and does not have the weakness above (in [2]), it (ii) requires defining a specific attribute and is not as broadly applicable to a marginal data distribution as our notion and (iii) requires a strong parametric assumption that the data is Gaussian (or close to Gaussian) whereas our approach is fully nonparametric.
>
> B. We will add a broader impact section to explicitly address the limitations of our privacy definition mentioned above (in [2]). Specifically, we will point our that while our approach will protect against an adversary inferring marginal data distributions, it may not protect specific individual attributes and another method, such as Zhang et al., 2022 may be more appropriate for such cases. We will explicitly state that our methodology does not protect against all cases and the data owner must consider the tradeoffs and the competitive risks associated with an adversary inferring specifics about the data distributions when choosing an appropriate methodology.

---

### Author Response · Authors · 2023-04-29
**New Revision Uploaded Addressing Reviewer Feedback**

We uploaded a new revision of our paper that includes all of the planned revisions mentioned in our responses to each reviewer. In particular, we included discussion of the additional papers referenced by Reviewer nErC, both in the related work and Discussion section, and added a Limitations subsection and Broader Impacts subsection to the Discussion section of the paper, addressing limitations and impacts from misuse both in regards to our proposed privacy notion and proposed generative model and how these relate to other approaches. We appreciate the thoughtful feedback from each of the reviewers and hope this revised version addresses the concerns raised.

---

### Decision · Action_Editors · 2023-05-26

**Recommendation:** Accept as is

**Comment:**

Reviewers KKXi and mZ5y found the paper strong and were happy with the authors' responses to their reviews.
Reviewer nErC was still worried that the privacy guarantee is weaker in nature than DP and attribute privacy, but this is openly discussed in Sec. 6.1 and the reviewer agreed that the paper meets TMLR acceptance criteria.

**Audience:**

All reviewers agree that at least some individuals would be interested in the findings of this paper.

**Claims And Evidence:**

All reviewers agree that the claims are supported by sufficient evidence.